# Hypothalamic neuronal circuits regulating hunger-induced taste modification

Ou Fu[1,2], Yuu Iwai[1], Masataka Narukawa[1], Ayako W. Ishikawa[3,4], Kentaro K. Ishii [1], Ken Murata[1], Yumiko Yoshimura[3,4], Kazushige Touhara [1,5], Takumi Misaka [1], Yasuhiko Minokoshi[2,4] & Ken-ichiro Nakajima [1,2,4]*

The gustatory system plays a critical role in sensing appetitive and aversive taste stimuli for evaluating food quality. Although taste preference is known to change depending on internal states such as hunger, a mechanistic insight remains unclear. Here, we examine the neuronal mechanisms regulating hunger-induced taste modification. Starved mice exhibit an increased preference for sweetness and tolerance for aversive taste. This hunger-induced taste modification is recapitulated by selective activation of orexigenic Agouti-related peptide (AgRP)-expressing neurons in the hypothalamus projecting to the lateral hypothalamus, but not to other regions. Glutamatergic, but not GABAergic, neurons in the lateral hypothalamus function as downstream neurons of AgRP neurons. Importantly, these neurons play a key role in modulating preferences for both appetitive and aversive tastes by using distinct pathways projecting to the lateral septum or the lateral habenula, respectively. Our results suggest that these hypothalamic circuits would be important for optimizing feeding behavior under fasting.

[1] Department of Applied Biological Chemistry, Graduate School of Agricultural and Life Sciences, The University of Tokyo, Bunkyo-ku, Tokyo 113-8657, Japan. [2] Division of Endocrinology and Metabolism, National Institute for Physiological Sciences, National Institutes of Natural Sciences, Okazaki 444-8585 Aichi, Japan. [3] Division of Visual Information Processing, National Institute for Physiological Sciences, National Institutes of Natural Sciences, Okazaki 444-8585 Aichi, Japan. [4] Department of Physiological Sciences, School of Life Science, SOKENDAI (The Graduate University for Advanced Studies), Okazaki 444-8585 Aichi, Japan. [5] International Research Center for Neurointelligence (WPI-IRCN), The University of Tokyo Institutes for Advanced Study, Tokyo 113-0033, Japan. *email: knakaj@nips.ac.jp

Central neural circuits for feeding behavior are highly complex and regulated by many factors such as internal state, emotion, and palatability of food[1]. Among them, the gustatory system plays a critical role in sensing appetitive and aversive taste stimuli for evaluating food quality[2–4]. For example, sweet taste and umami taste are symbols of a calorie-rich diet containing sugars and amino acids, respectively. Thus, animals prefer and consume them as much as possible to escape hunger. By contrast, sour and bitter tastes are thought to be a symbol of spoiled or poisonous food. Thus, animals refuse to eat them for their own safety.

Taste preference and sensitivity are the most important determinants of food evaluation. Importantly, such criteria are not always constant and often change depending on internal states such as hunger and satiety. Recent evidence indicates that hunger induces increased sweet taste preference and/or sensitivity in various species from fruit flies to humans[5,6]. Electrophysiological recordings of various neurons in several brain areas, such as the amygdala, the orbital frontal cortex, and the hypothalamus, in mice and in monkeys, have indicated the existence of neurons that can respond to taste stimuli in a state (hunger/satiety)-dependent manner[7–9]. However, the key neuronal pathway(s) responsible for hunger-induced taste modification remain(s) unknown.

Neural circuits for hunger have been extensively investigated, and the importance of Agouti-related peptide (AgRP)-expressing neurons under hunger conditions has been highlighted[10]. AgRP neurons are localized in the arcuate nucleus (ARC) of the hypothalamus and are normally activated by physiological hunger in order to trigger feeding behavior[11]. Either chemogenetic or optogenetic activation of AgRP neurons is sufficient to induce acute food intake even under satiated conditions[12,13]. AgRP neurons send inhibitory inputs to multiple brain areas, including both the intra- and extra-hypothalamus[14]. Although some of these areas directly induce feeding behavior, the others play roles in affecting feeding-unrelated behaviors (e.g., mating and pain)[15,16]. By contrast, the mechanism by which AgRP-neuron originating hunger circuits contribute to taste modification is unknown.

To address this issue, we investigated the effects of AgRP neurons on hunger-induced taste modification by using chemogenetic and optogenetic approaches to minimize potential hormonal effects on taste sensations.

We demonstrate that physiological hunger affects preferences to both appetitive and aversive tastes and these effects are recapitulated by artificial activation of the lateral hypothalamus (LHA)-projecting AgRP neurons. Next, we show that glutamatergic neurons, but not GABAergic neurons, in the LHA (Vglut2[LHA] neurons) are a downstream functional module of AgRP neurons. Importantly, these neurons play a key role in modifying preferences for both appetitive and aversive tastes. Interestingly, two distinct neuronal pathways starting from the LHA to the lateral septum (LS) or to the lateral habenula (LHb) contribute to the modulation of appetitive and aversive taste preferences, respectively. These findings clarify the existence of hypothalamic neuronal circuits (AgRP[ARC]→Vglut2[LHA]→LS and AgRP[ARC]→Vglut2[LHA]→LHb) regulating hunger-induced taste modification.

## Results

**AgRP neurons modulate sweet and bitter taste preferences**. As previously reported, taste preferences change under physiological hunger conditions in humans and fruit flies[3,4]. Thus, we first tested whether a similar phenomenon is observed in mice. The brief access taste test is commonly used to evaluate taste preference in mice (Fig. 1a)[17–19]. Counting of the number of licks of a taste solution in a short period (10 s) allows us to measure taste preference without post-ingestive effects. We thus performed the brief access taste test by using wild-type (WT) mice under fed and overnight fasted conditions (Fig. 1a–c). While the number of licks of the high concentration of sucrose solution (300 mM) was comparable between the fed and fasted groups, fasted mice exhibited a stronger preference toward moderate concentrations of sucrose (100 mM) (Fig. 1b).

Next, we tested sensitivity to bitter taste by presenting a sugar solution mixed with various concentrations of bitter substances. As shown in Fig.1c, the mixture of a bitter tastant (denatonium) suppressed the licking response to 500 mM sucrose in fed mice in a dose-dependent manner (Fig. 1c, "Fed"). Interestingly, during starvation, there was a reduction in bitter sensitivity, as indicated by a rightward shift in the dose–response curve for licking inhibition as a function of denatonium concentration (Fig. 1c, "Fasted"). These results demonstrate that starvation modifies appetitive and aversive tastes in the opposite direction.

Because AgRP neurons in the ARC of the hypothalamus are activated by physiological hunger in order to trigger acute increase in food intake[10], we tested whether activation of these neurons induces taste modification as observed under overnight fasting. For this purpose, we selectively expressed hM3Dq, which is an excitatory designer receptor exclusively activated by designer drugs (DREADD[20]), in AgRP neurons. An adeno-associated virus (AAV) encoding Cre-dependent hM3Dq (AAV-hSyn-DIO-hM3Dq-mCherry) was bilaterally injected into the ARC of AgRP-ires-Cre knock-in mice expressing Cre recombinase exclusively in AgRP neurons (hereafter, AgRP-hM3Dq mice) (Fig. 1d). After systemic injection (1.0 mg/kg intraperitoneally (i.p.)) of the DREADD agonist, clozapine N-oxide (CNO), which is otherwise pharmacologically inert, into AgRP-hM3Dq mice, we observed robust c-fos expression in the hM3Dq-mCherry-expressing AgRP neurons (Fig. 1e; Supplementary Fig. 1A). Importantly, a dramatic increase in food intake was also observed in AgRP-hM3Dq mice after CNO injection as in the case of overnight-fasted mice (Fig. 1f; Supplementary Fig. 4G). By contrast, mice injected with the control AAV encoding Cre-dependent mCherry showed little c-fos expression and no change in food intake after CNO treatment (Supplementary Fig. 1B, C). We then evaluated whether chemogenetic activation of AgRP neurons affects taste preference. Importantly, activation of AgRP neurons led to an increase in the relative lick ratio of the sucrose solution (100 mM) (Fig. 1g). By contrast, such change was not observed in mice injected with the control AAV encoding Cre-dependent mCherry (Supplementary Fig. 1D). As this phenotype was also observed in AgRP-hM3Dq mice treated with the non-calorie sweetener, sucralose, the enhancement is likely due to the sweet taste itself and not calorie content (Supplementary Fig. 2A). We next evaluated behavioral sensitivity to aversive taste in the same AgRP-hM3Dq mice. The lick ratio of the denatonium solution (mixed with sucrose) decreased in saline-treated mice in a dose-dependent manner (Fig. 1h "saline"). By contrast, CNO treatment induced a reduction in aversive response to bitter taste, as indicated by a rightward shift in the dose–response curve for licking inhibition as a function of denatonium concentration (Fig. 1h, "CNO"). These phenotypes were quite similar to those observed in fasted mice (Fig. 1b, c).

To determine if the decrease in bitter taste sensitivity was due to a "masking" effect of the increased preference to the sucrose in the mixture solution, we performed a brief access test by using a bitter solution without sucrose. For this purpose, AgRP-hM3Dq mice were placed on a 23-h water-deprivation schedule to increase the motivation to lick. Similar to the use of the bitter–sweet mixture solution (Fig. 1h), AgRP-hM3Dq mice

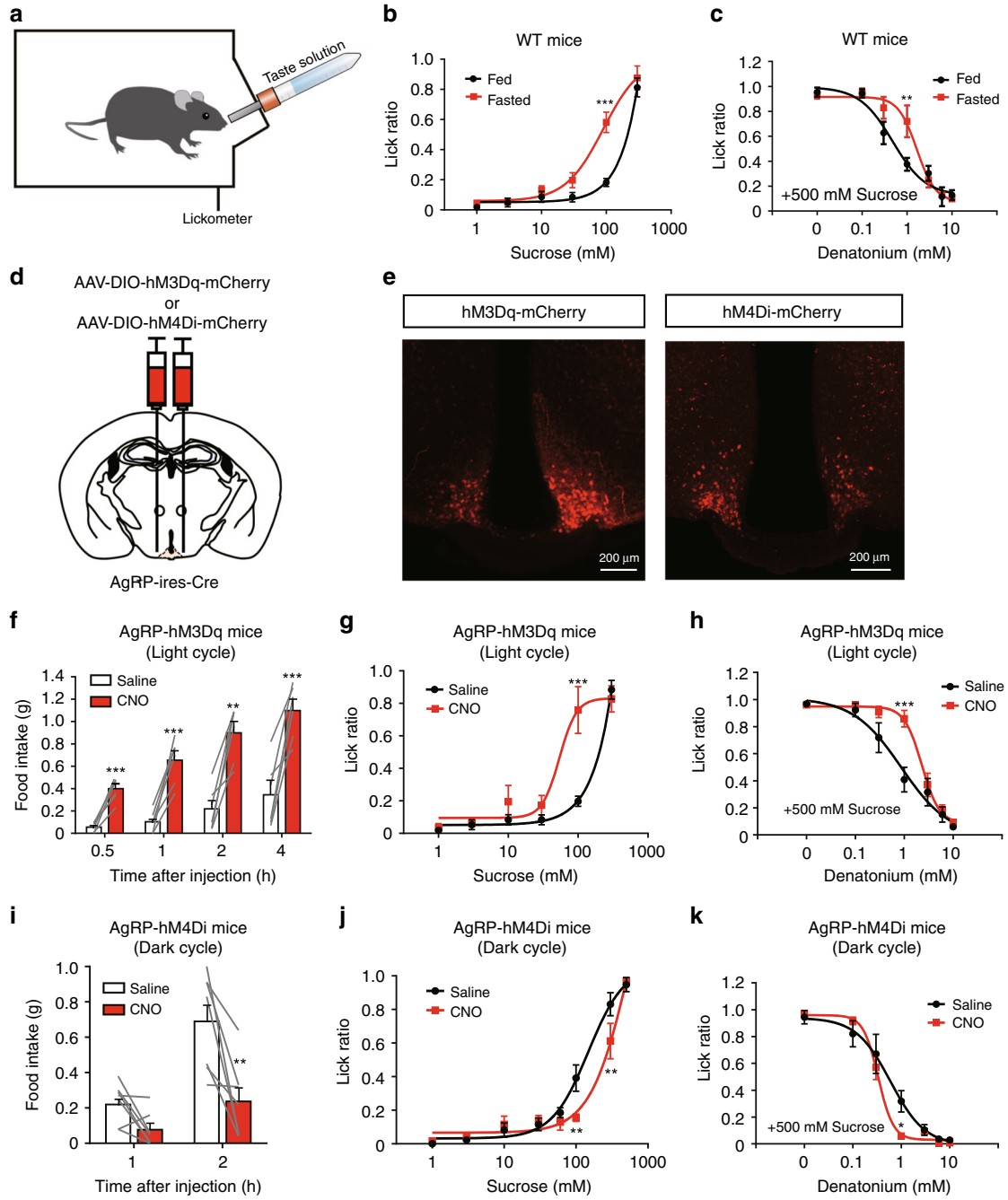

**Fig. 1** Chemogenetic activation of AgRP neurons induces changes in taste preference. **a** Schematic image of the brief access taste test. The number of licks is measured during 10 s from the first lick. **b**, **c** Sweet (**b**) or bitter (**c**) taste preferences in fed or fasted mice. Sucrose or denatonium–sucrose solutions were presented to fed or 23-h-fasted C57BL/6J WT mice. $n = 6$, $F = 17.81$, and $P = 9.4 \times 10^{-5}$ in **b** and $n = 6$, $F = 4.14$, and $P = 0.045$ in **c**, two-way ANOVA with Bonferroni post hoc test. **d** Bilateral injection of AAV encoding Cre-dependent hM3Dq-mCherry or hM4Di-mCherry into the arcuate nucleus (ARC) of AgRP-ires-Cre mouse. **e** Representative image showing hM3Dq-mCherry-expressing AgRP neurons (left) in the AgRP-hM3Dq mouse and hM4Di-mCherry-expressing AgRP neurons (right) in the AgRP-hM4Di mouse. **f** Chemogenetic activation of AgRP neurons led to acute food intake in AgRP-hM3Dq mice during the light period. $n = 6$, paired Student's $t$ test. **g**, **h** Brief access taste tests for sweet (**g**) or bitter (**h**) measured in AgRP-hM3Dq mice treated with saline or CNO (1.0 mg/kg i.p.) during the light cycle. $n = 6$, $F = 8.783$, and $P = 0.0045$ in **g** and $n = 6$, $F = 7.929$, and $P = 0.0064$ in **h**, two-way ANOVA with Bonferroni post hoc test. **i** Chemogenetic inhibition of AgRP neurons led to a reduction of food intake in AgRP-hM4Di mice during the dark cycle. $n = 7$, paired Student's $t$ test. **j**, **k** Brief access taste tests for sweet (**j**) or bitter (**k**) measured in AgRP-hM4Di mice treated with saline or CNO (1.0 mg/kg i.p.) during the dark cycle. $n = 7$, $F = 4.748$, and $P = 0.032$ in **j** and $n = 7$, $F = 4.761$, and $P = 0.032$ in **k**, two-way ANOVA with Bonferroni post hoc test. The experiments were carried out with 8- to 16-week-old male mice. Data are given as means ± SEM. $*P < 0.05$, $**P < 0.01$, $***P < 0.001$

showed more tolerance to the denatonium solution after CNO treatment compared with the saline-injected group (Supplementary Fig. 2B), suggesting that bitter sensitivity decreases during activation of AgRP neurons independent of an increased sucrose preference. Importantly, chemogenetic activation of AgRP neurons led to a decrease in sour taste sensitivity. This tolerance is similar to that observed in the overnight-fasted mice (Supplementary Fig. 2C). These results suggest that AgRP-neuron-induced taste modification occurred for aversive tastes in general, and that this response was not exclusive to bitter taste.

We next examined whether suppression of AgRP neurons affects taste preference in mice. We injected AAV-expressing Cre-dependent inhibitory DREADD (AAV-hSyn-DIO-hM4Di-mCherry) into the ARC of AgRP-ires-Cre mice (hereafter called AgRP-hM4Di mice). AgRP-hM4Di mice treated with saline consumed large amounts of food in the initial 2 h during the dark cycle (Fig. 1i). The feeding pattern is similar to that observed in the case of chemogenetic activation of AgRP neurons in the light cycle (Fig. 1f). In contrast, AgRP-hM4Di mice treated with CNO exhibited significantly decreased food intake for the initial 2 h of the dark cycle (Fig. 1i) as previously reported[13]. Interestingly, the brief access taste test demonstrated that chemogenetic inhibition of AgRP neurons reverses either appetitive or aversive taste preference under physiological hunger conditions (Fig. 1j, k). Collectively, these results strongly suggest that hunger-induced taste modification is regulated by the activity of AgRP neurons.

**LHA-projecting AgRP neurons modulate sweet and bitter tastes.** Gustatory nerve recording experiments by using AgRP-hM3Dq mice showed no difference in the responses to sweet and bitter tastes in the presence or absence of CNO (Supplementary Fig. 3A–C). These results indicate that AgRP neurons do not affect the peripheral taste system but rather affect higher brain regions. As AgRP neurons project to various brain areas including both the intra- and extra-hypothalamus[14], there is the possibility that one or more sites among these areas contribute to AgRP-neuron-induced taste modification. To determine which projection area of AgRP neurons regulates taste preferences, we first visualized the axon terminals of AgRP neurons by injecting the anterograde tracer AAV (AAV-hEF1a-DIO-synaptophysin-mCherry) into AgRP-ires-Cre mice (Supplementary Fig. 4A). We found a dense innervation in the paraventricular nucleus of the hypothalamus (PVH), the LHA, and the central nucleus of the amygdala (CEA) as reported previously (Supplementary Fig. 4B)[14]. To selectively activate individual axon terminals, we performed optogenetic experiments. Similar to the DREADD experiment, we bilaterally delivered the AAV-encoding Cre-dependent light-sensitive channel rhodopsin-2 (AAV-FLEX-rev-ChR2-tdTomato) into the ARC of AgRP-ires-Cre mice (Supplementary Fig. 4A, hereafter, AgRP-ChR2 mice), and optical fibers were placed slightly above the ARC (soma) or axon terminals (Fig. 2a, c, f, i, l).

To evaluate the activation of ChR2 in AgRP neurons, we recorded the action potentials from the ChR2-expressing AgRP neurons in acute hypothalamic slices. Blue-flash light applied to the recorded neurons induced action potentials in all tested ChR2-expressing AgRP neurons ($n = 7$) in the cell-attached mode (Supplementary Fig. 4D). The firing rate for the recorded neurons was significantly increased during photostimulation ($9.3 \pm 1.44$ Hz), compared with that before the stimulation ($0.88 \pm 0.45$ Hz). In addition, we recorded the membrane potentials from ChR2-expressing AgRP neurons with a whole-cell recording in the current-clamp mode in the presence of TTX. The membrane potentials were depolarized during the photostimulation (Supplementary Fig. 4E). The peak of the depolarization evoked by the first pulse of photostimulation at the most

effective stimulation sites was $-21.34 \pm 3.61$ mV ($n = 11$), which was significantly higher than the resting-membrane potential before photostimulation ($-45.27 \pm -2.25$ mV).

These results showed that ChR2-expressing AgRP neurons were efficiently activated with light in our preparation.

As reported previously, in vivo photostimulation of the ARC of AgRP-ChR2 mice significantly increased food intake acutely with a corresponding increase in *c-fos* expression as observed in the case of overnight-fasted mice (Supplementary Fig. 4C, F, G)[12]. By contrast, the number of water licks did not change within 10 min of the optogenetic activation of AgRP neurons (Supplementary Fig. 4H). These results indicate that activation of AgRP neurons induced appetite but not thirst. Next, we selectively activated the axon terminals of AgRP neurons by optogenetically stimulating the projection area of AgRP neurons in AgRP-ChR2 mice. While activation of either PVH-projecting or LHA-projecting AgRP neurons led to an increase in food intake, activation of CEA-projecting AgRP neurons did not induce an increase in food intake as reported previously (Supplementary Fig. 4I)[14]. These results confirmed that the orexigenic activity of AgRP neurons occurs in multiple, but not all, projection areas of AgRP neurons.

We then evaluated the role of AgRP neurons and their individual projection areas on taste preference by using brief access taste tests in AgRP-ChR2 mice in the presence of photostimulation. As expected, optogenetic activation of the soma of AgRP neurons leads to a similar phenotype in taste modification as observed in overnight-fasted mice (Figs. 1b, c and 2c–e). Next, we selectively activated each axon terminal of the AgRP neurons and found that only selective activation of the axon terminals of LHA-projecting AgRP neurons is able to induce the enhancement of sweet taste preference (Fig. 2g, j, m). Furthermore, such enhancement was observed when using the non-calorie sweetener, sucralose instead of sucrose, indicating that the modification is not calorie-dependent, but sweet taste-dependent (Supplementary Fig. 5B, H).

Interestingly, selective activation of LHA-projecting AgRP neurons is also able to induce tolerance to aversive taste such as bitter and sour tastes (Fig. 2k; Supplementary Fig. 5I). By contrast, this change was not observed when stimulating PVH-projecting AgRP neurons (Fig. 2f–h) or CEA-projecting AgRP neurons (Fig. 2l–n). Other reported projection areas of AgRP neurons did not induce taste modification (Supplementary Fig. 6)[14]. These results strongly indicate that LHA-projecting AgRP neurons play a critical role in modifying taste-guided licking behaviors under hunger conditions.

**LHA-projecting AgRP neurons do not change sweet sensitivity.** We next investigated whether the increased lick rate for moderately concentrated sucrose solutions during optogenetic activation of LHA-projecting AgRP neurons (Fig. 2j) is due to change in sweet taste sensitivity. As conditioned taste aversion test (CTA) is commonly used to accurately measure perceived taste thresholds for appetitive taste qualities such as sweet[17], we performed the CTA test combined with optogenetic experiments. During CTA learning, we injected LiCl into AgRP-ChR2 mice that have bilateral optic fibers above the LHA (Fig. 3a, b) soon after sweet taste presentation. After acquisition of CTA, the AgRP-ChR2 mice exhibited a strong aversion to 300 mM sucrose solution (Fig. 3c). Importantly, optogenetic activation of LHA-projecting AgRP neurons did not affect the dose-dependent inhibitory response toward sucrose (Fig. 3d). These results suggest that LHA-projecting AgRP neurons do not contribute to a lower sweet taste threshold, implying that they enhance the hedonic aspect of sweet taste[21].

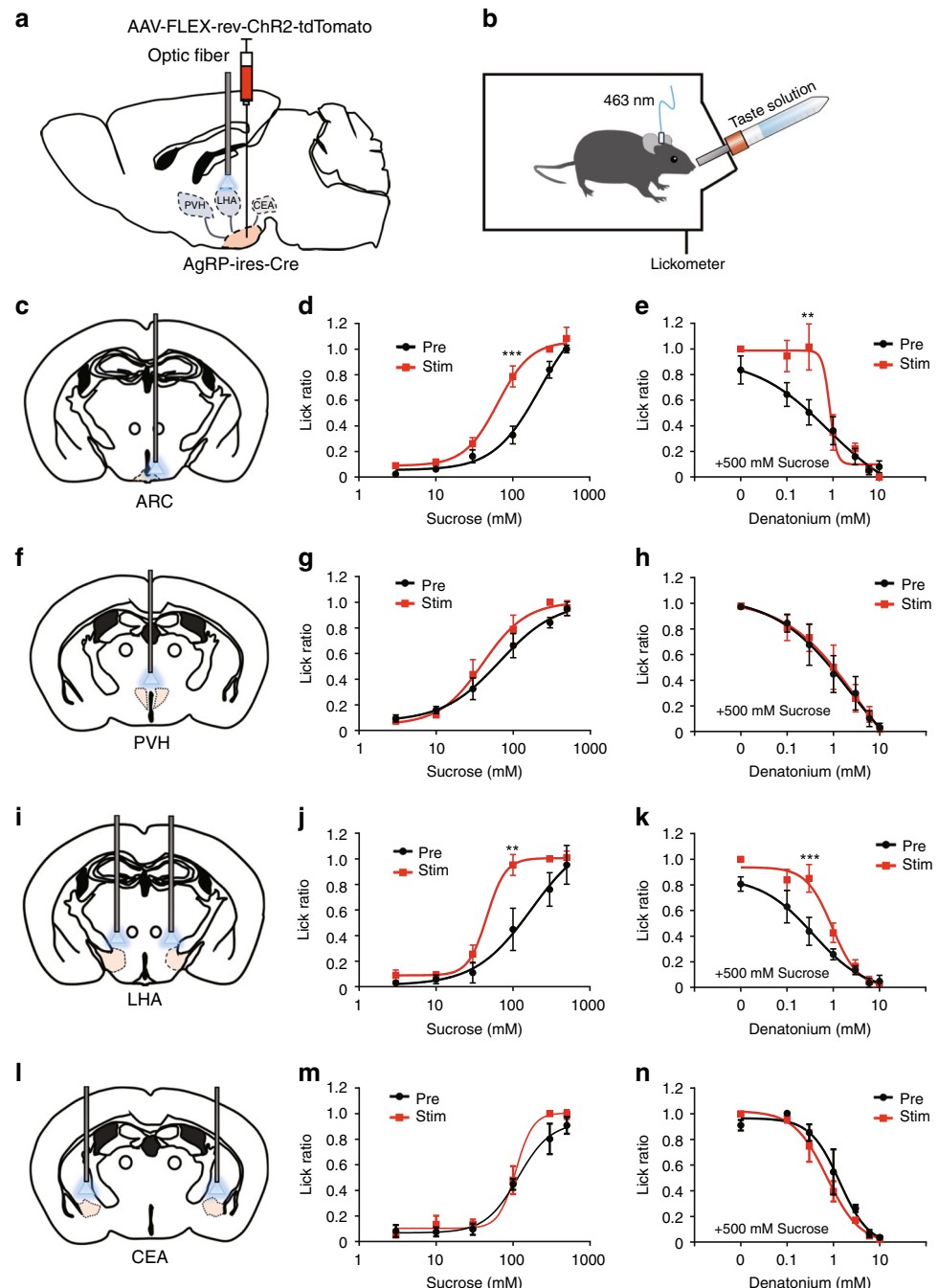

**Fig. 2** Optogenetic activation of LHA-projecting AgRP neurons modulates taste preferences. **a** AAV-FLEX-rev-ChR2-tdTomato was bilaterally injected into the ARC of AgRP-ires-Cre mice, and optical fibers were placed above the projection regions of AgRP neurons. **b** Schematic image of the brief access taste test during in vivo optogenetic activation of AgRP axon terminals in AgRP-ChR2 mice. **c–e** Brief access taste tests toward sweet (**d**) or bitter (**e**) solutions during photostimulation of the soma of AgRP$^{ARC}$ neurons. $n = 11$, $F = 19.41$, and $P = 2.8 \times 10^{-5}$ in **d** and $n = 11$, $F = 6.926$, and $P = 0.01$ in **e**, two-way ANOVA with Bonferroni post hoc test. **f–n** Brief access taste tests for sweet and bitter solutions when exclusively activating PVH-projecting (**f–h**), LHA-projecting (**i–k**), and CEA-projecting (**l–n**) AgRP neurons, respectively. $n = 6$, $F = 2.263$, and $P = 0.138$ in **g**, $n = 6$, $F = 0.2445$, and $P = 0.87$ in **h**, $n = 5$, $F = 11.42$, and $P = 0.0015$ in **j**, $n = 5$, $F = 13.88$, and $P = 0.0005$ in **k**, $n = 5$, $F = 3.174$, and $P = 0.081$ in **m**, and $n = 5$, $F = 1.729$, and $P = 0.194$ in **n**, two-way ANOVA with Bonferroni post hoc test. All experiments were carried out with 10- to 16-week-old male mice. Data are given as means ± SEM. *$P < 0.05$, **$P < 0.01$, ***$P < 0.001$, as compared with the corresponding control group

**Vglut2$^{LHA}$ neurons are a downstream target of AgRP neurons.** AgRP neurons are GABAergic and thus send inhibitory input into their downstream targets[22]. In the LHA, there are two types of neurons: GABAergic and glutamatergic. While artificial inhibition of the vesicular GABA transporter (Vgat)-expressing neurons in the LHA led to a decrease in food intake, artificial inhibition of the vesicular glutamate transporter2 (Vglut2)-expressing neurons in

the LHA led to an increase in food intake[23,24]. These results prompted us to investigate a possible connection between AgRP neurons and Vglut2$^{LHA}$ neurons. To confirm this connection, we first injected AAV-Flex-tdTomato into the LHA of Vglut2-ires-Cre mice for labeling (Fig. 4a). We observed that tdTomato-positive Vglut2 neurons and the axon terminals of AgRP neurons are in close proximity to each other in the LHA (Fig. 4a).

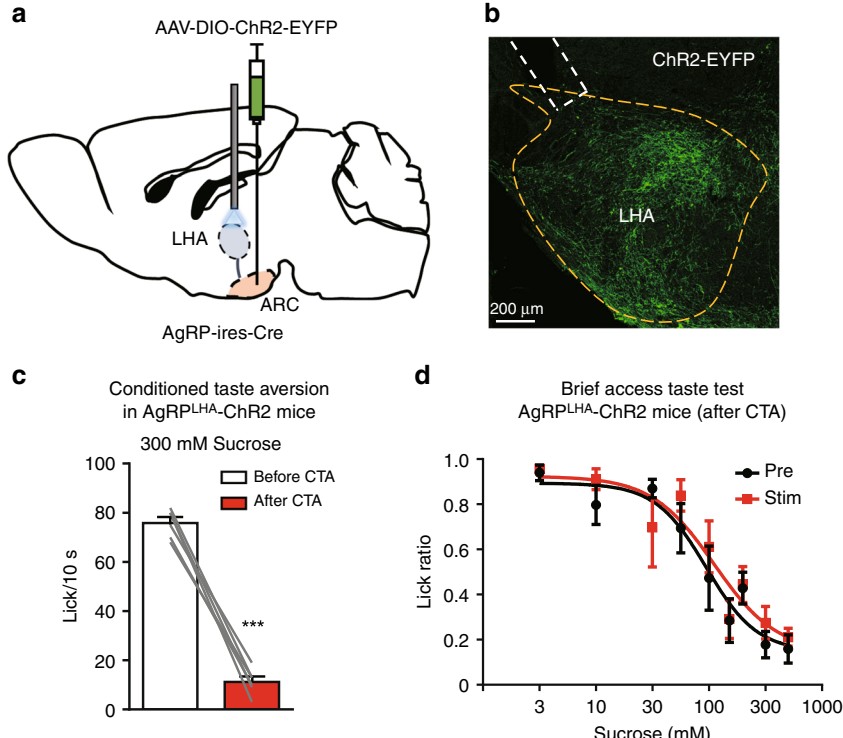

**Fig. 3** LHA-projecting AgRP neurons do not affect sweet taste sensitivity after CTA. **a** Schematic image of the injection of AAV-DIO-ChR2-EYFP into the ARC with bilateral optic fiber insertion above the LHA for photostimulation in the AgRP-ChR2-EYFP mouse. **b** Representative image showing ChR2-EYFP expression at axon terminals of AgRP neurons in the LHA and the approximate placement of an optic fiber (dashed lines). **c** Licks for 300 mM sucrose in 10 s before and after CTA conditioning. $n = 6$, paired Student's $t$ test. **d** Brief access taste tests for sucrose solution in the presence or absence of optogenetic activation of LHA-projecting AgRP neurons after CTA conditioning. $n = 6$, $F = 1.272$, and $P = 0.262$, two-way ANOVA with Bonferroni post hoc test. All experiments were carried out with 10- to 16-week-old male mice. Data are given as means ± SEM. ***$P < 0.001$

To further confirm the connection between AgRP neurons and Vglut2$^{LHA}$ neurons, we next carried out monosynaptic rabies-tracing experiments as previously reported (Fig. 4b)[25]. We injected two AAVs encoding TVA or RG into the LHA of Vglut2-ires-Cre mice. Three weeks after AAV injection, recombinant rabies lacking glycoprotein (Rabies ΔG-GFP + EnvA) was introduced into the same target as the initial AAV injection. Retrograde trans-synaptic labeling from Vglut2$^{LHA}$ neurons indicated that direct synaptic connections exist between AgRP neurons and Vglut2$^{LHA}$ neurons (Fig. 4c). We also observed that some GFP-labeled neurons did not overlap with the AgRP staining, indicating that parts of the Vglut2$^{LHA}$ neurons receive input from other neurons in the ARC (Fig. 4c). Collectively, these data indicate that LHA-projecting AgRP neurons have synaptic connections to Vglut2$^{LHA}$ neurons.

**Vglut2$^{LHA}$ neurons modulate taste preferences.** We next investigated whether Vglut2$^{LHA}$ neurons contribute to AgRP-neuron-induced taste modification. To selectively inhibit or activate Vglut2$^{LHA}$ neurons, we injected Vglut2-ires-Cre mice with the AAV-encoding inhibitory DREADD (AAV-hSyn-DIO-hM4Di-mCherry, Vglut2$^{LHA}$-hM4Di mice) or excitatory DREADD (AAV-hSyn-DIO-hM3Dq-mCherry, Vglut2$^{LHA}$-hM3Dq mice), respectively (Fig. 5a–c). Chemogenetic inhibition of Vglut2$^{LHA}$ neurons (1.0 mg/kg i.p. CNO injection) in Vglut2$^{LHA}$-hM4Di mice led to an increase in food intake during 1 h after CNO injection (Fig. 5d). Brief access taste tests showed that CNO-treated Vglut2$^{LHA}$-hM4Di mice have a stronger preference for sweet tastes and exhibit tolerance to bitter tastes, as in the case of overnight-fasted mice (Figs. 5e, f and 1b, c).

These results suggest that Vglut2$^{LHA}$ neurons function as a downstream module of LHA-projecting AgRP neurons. Conversely, chemogenetic activation of Vglut2$^{LHA}$ neurons strongly suppressed refeeding responses in starved mice within 1 h (0.66 g ± 0.03 g in the saline-treated control group ($n = 6$ mice) versus 0.10 g ± 0.03 g in the CNO-treated group ($n = 6$ mice), Fig. 5g). We then evaluated the effects of the activation of Vglut2$^{LHA}$ neurons on hunger-induced taste modification. Strikingly, overnight-fasted Vglut2$^{LHA}$-hM3Dq mice treated with CNO largely lost the hunger-induced change in lick responses toward either sweet or bitter tastes (Fig. 5, h, i). Collectively, these results show that Vglut2$^{LHA}$ neurons play a key role in hunger-induced taste modification.

**GABAergic LHA neurons do not modulate taste preferences.** We next investigated the role of inhibitory GABAergic neurons in the LHA on hunger-induced taste modification. Similar to the case of Vglut2$^{LHA}$ neurons, we carried out monosynaptic rabies tracing with GAD2-Cre mice that selectively express Cre in GABAergic neurons. The results indicated that parts of AgRP neurons connect to GABAergic neurons in the LHA (Supplementary Fig. 7A). Considering the fact that AgRP neurons are inhibitory neurons, we next chemogenetically inhibited GABAergic neurons to determine their effects on taste preference. As previously reported, chemogenetic inhibition of GABAergic LHA neurons leads to a decrease in food intake during the dark cycle[23] (Supplementary Fig. 7B, C). In clear contrast to Vglut2$^{LHA}$ neurons (Fig. 5), taste preference did not change due to inhibition of GABAergic LHA neurons in fed mice (Supplementary Fig. 7D, E). These results suggest that the contribution of

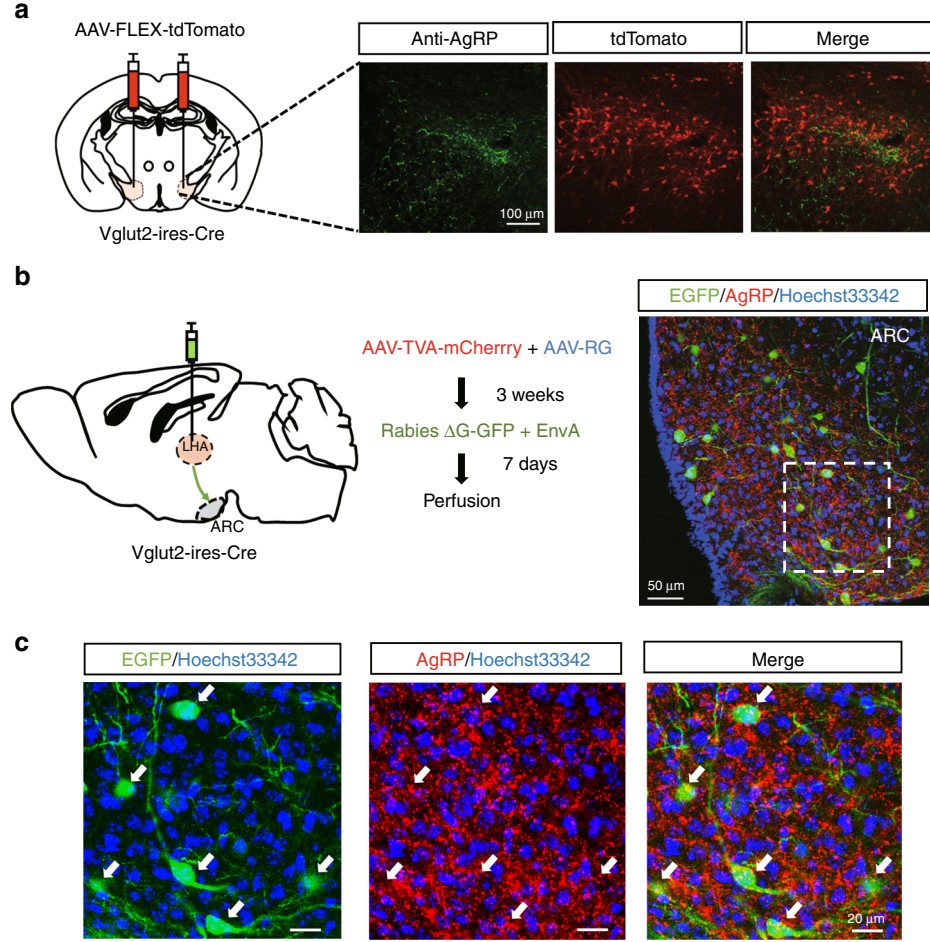

**Fig. 4** AgRP neurons connect to parts of the glutamatergic Vglut2-expressing neurons in the LHA. **a** Representative immunohistochemical image of a coronal section of AgRP axon terminals (green) and Vglut2 neurons (red) in the LHA. Vglut2 neurons were labeled with mCherry by injection of AAV-FLEX-tdTomato in the LHA of Vglut2-ires-Cre mice. **b** Schematic image of the monosynaptic rabies tracing of Vglut2 neurons in the LHA (left) and a brain section after monosynaptic-tracing neurons in the ARC that are synaptically connected to Vglut2$^{LHA}$ neurons in the ARC (right). **c** Enlarged view of ARC neurons that are monosynaptically connected to Vglut2$^{LHA}$ neurons (green) and AgRP immunostaining (red) in the ARC. Arrows show parts of AgRP neurons monosynaptically connected to Vglut2$^{LHA}$ neurons

GABAergic LHA neurons to taste preferences is much smaller than that induced by glutamatergic LHA neurons.

**Two pathways of Vglut2$^{LHA}$ neurons on the taste modification**. Vglut2$^{LHA}$ neurons are known to project to various brain areas to regulate diverse behaviors including feeding[23,24,26]. To map the projection area of Vglut2$^{LHA}$ neurons, we expressed hM4Di-mCherry, which tends to accumulate in the presynaptic area in Vglut2$^{LHA}$ neurons since it is an anterograde tracer[25,27] (Fig. 6a). We observed dense projections in the following three areas: the LS, the anterodorsal thalamus (AD), and the LHb (Fig. 6b).

To clarify the role of these projection areas on taste modification, we performed a pathway-selective chemogenetic inhibition study. Microinjections of CNO in the hM4Di-expressing axon terminals of Vglut2$^{LHA}$ neurons (Supplementary Fig. 8A-C) allowed us to selectively inhibit the corresponding neuronal pathway[26,27].

Interestingly, inhibition of LS-projecting Vglut2$^{LHA}$ neurons led to the enhancement of sweet taste preference; however, the tolerance to bitter taste was not changed in the same mice (Fig. 6c-e). By contrast, inhibition of LHb-projecting Vglut2$^{LHA}$ neurons did not affect sweet taste preference; however, it did lead to a selective tolerance to bitter taste (Fig. 6f-h). Licking responses toward either sweet or bitter taste were not changed

when inhibiting AD-projecting Vglut2$^{LHA}$ neurons (Fig. 6i-k). Collectively, these data indicate that sweet and bitter taste preferences are modified depending on the activities of LS-projecting Vglut2$^{LHA}$ neurons or LHb-projecting Vglut2$^{LHA}$ neurons, respectively.

**Distinct glutamatergic LHA neurons project to the LS and LHb**. As LS-projecting Vglut2$^{LHA}$ neurons and LHb-projecting Vglut2$^{LHA}$ neurons separately modulate the preference for sweet and bitter tastes, we evaluated whether LS-projecting Vglut2$^{LHA}$ neurons and LHb-projecting Vglut2$^{LHA}$ neurons are the same. For this purpose, we injected retrobeads with two different colors into the LS (red) and the LHb (green) in the same C57/BL6J mice (Fig. 7a; Supplementary Fig. 8D, E). One week after injection, we checked the distribution of the retrobeads in the LHA (Fig. 7b, c). Interestingly, the majority of retrobead-positive LHA neurons (97.4% ± 0.4%, $n = 2$ mice) were singly labeled along the anterior–posterior axis (Fig. 7d). These results strongly suggest that the LS and the LHb are innervated by distinct neuronal populations in the LHA. We further evaluated whether LS-projecting and LHb-projecting LHA neurons are Vglut2 neurons. For this purpose, we injected an AAV-encoding Cre-dependent green fluorescent protein (GFP) into the LHA of Vglut2-ires-Cre neurons together with red retrobeads into the LS

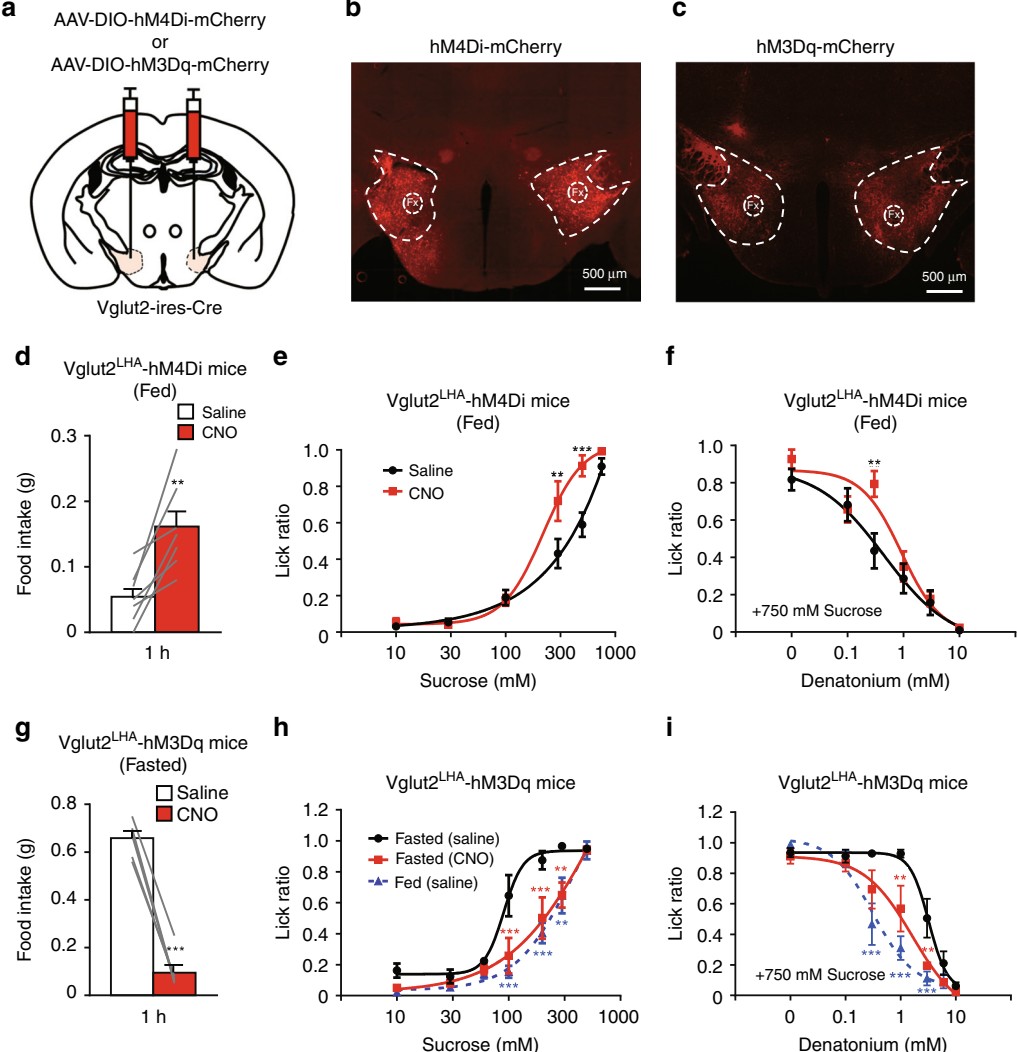

**Fig. 5** Vglut2 neurons are necessary for hunger-induced modulation of taste preference. **a** Bilateral injection of AAV-DIO-hM4Di-mCherry or AAV-DIO-hM3Dq-mCherry into the LHA of Vglut2-ires-Cre mice. **b–c** Representative images of hM4Di-mCherry (**b**) and hM3Dq-mCherry (**c**) expression in Vglut2$^{LHA}$ neurons. Fx, fornix. **d** Chemogenetic inhibition (CNO 1.0 mg/kg i.p.) of Vglut2 neurons in the LHA promotes food intake in Vglut2-hM4Di mice within 1 h $n = 7$, paired Student's $t$ test. **e–f** Brief access tests with sweet (**e**) and bitter (**f**) taste solutions in Vglut2$^{LHA}$-hM4Di mice in the presence or absence of CNO (1.0 mg/kg i.p.). $n = 6$, $F = 11.99$, and $P = 0.001$ in **e** and $n = 6$, $F = 5.37$, and $P = 0.023$ in **f**, two-way ANOVA with Bonferroni post hoc test. **g** Chemogenetic activation (CNO 1.0 mg/kg i.p.) of Vglut2 neurons in the LHA led to a decrease in 1-h food intake in 23-h-fasted Vglut2$^{LHA}$-hM3Dq mice. **h–i** Brief access test with sweet (**h**) and bitter (**i**) taste solutions in Vglut2$^{LHA}$-hM3Dq mice under fed or 23-h-fasted conditions in the presence or absence of CNO (1.0 mg/kg i.p.). $n = 6$, $F = 21.77$, and $P = 1.5 \times 10^{-8}$ in **h** and $n = 6$, $F = 19.51$, and $P = 1.1 \times 10^{-7}$ in **i**, two-way ANOVA with Bonferroni post hoc test. All experiments were carried out with 10- to 16-week-old male mice. Data are given as means ± SEM. *$P < 0.05$, **$P < 0.01$, ***$P < 0.001$, as compared with the saline group (**d–g**) and with the fasted (saline) group (**h–i**).

or the LHb (Supplementary Fig. 9A, C). We observed an overlap of GFP and the red retrobeads in the LHA neurons (Supplementary Fig. 9B, D). These data together with the behavioral experiment results indicate that different groups of Vglut2$^{LHA}$ neurons connected to the LS or to the LHb regulate appetitive and aversive taste preferences, respectively.

**Fasting affects *c-fos* expression in the LS and the LHb**. We finally examined the effects of hunger on taste-induced *c-fos* expression in the LS and the LHb. In the LS, basal *c-fos* expression was significantly decreased by sweet taste compared with the control group (Fig. 8a, c). Interestingly, this sweet- induced *c-fos* reduction was dramatically enhanced by overnight fasting (51.0% ± 1.6% reduction, $n = 3$ mice and 84.6% ± 1.2% reduction, $n = 5$ mice, respectively, compared with the control group, $n = 3$ mice) (Fig. 8c). Importantly, such a large decrease was not

observed when bitter taste was provided under hunger conditions (17.7% ± 4.2%, $n = 3$ mice) (Fig. 8c).

In the LHb, the number of *c-fos*-positive cells was small in the control group (Fig. 8b, d). While either sweet or bitter taste induced *c-fos* expression in the LHb in fed mice, fasting selectively suppressed bitter-induced *c-fos* expression (53.0% ± 8.1% decrease, $n = 3$ mice, compared with the non-fasted bitter group, $n = 3$ mice) (Fig. 8b, d). Collectively, these results suggest that fasting differentially affects taste-induced *c-fos* expression in the LS and the LHb, respectively.

## Discussion

In this study, we showed that hunger enhances sweet taste preference and decreases aversive taste sensitivity (Fig. 1b, c). Considering the basic role of the gustatory system in the wild environment, this bidirectional modulation is likely needed to

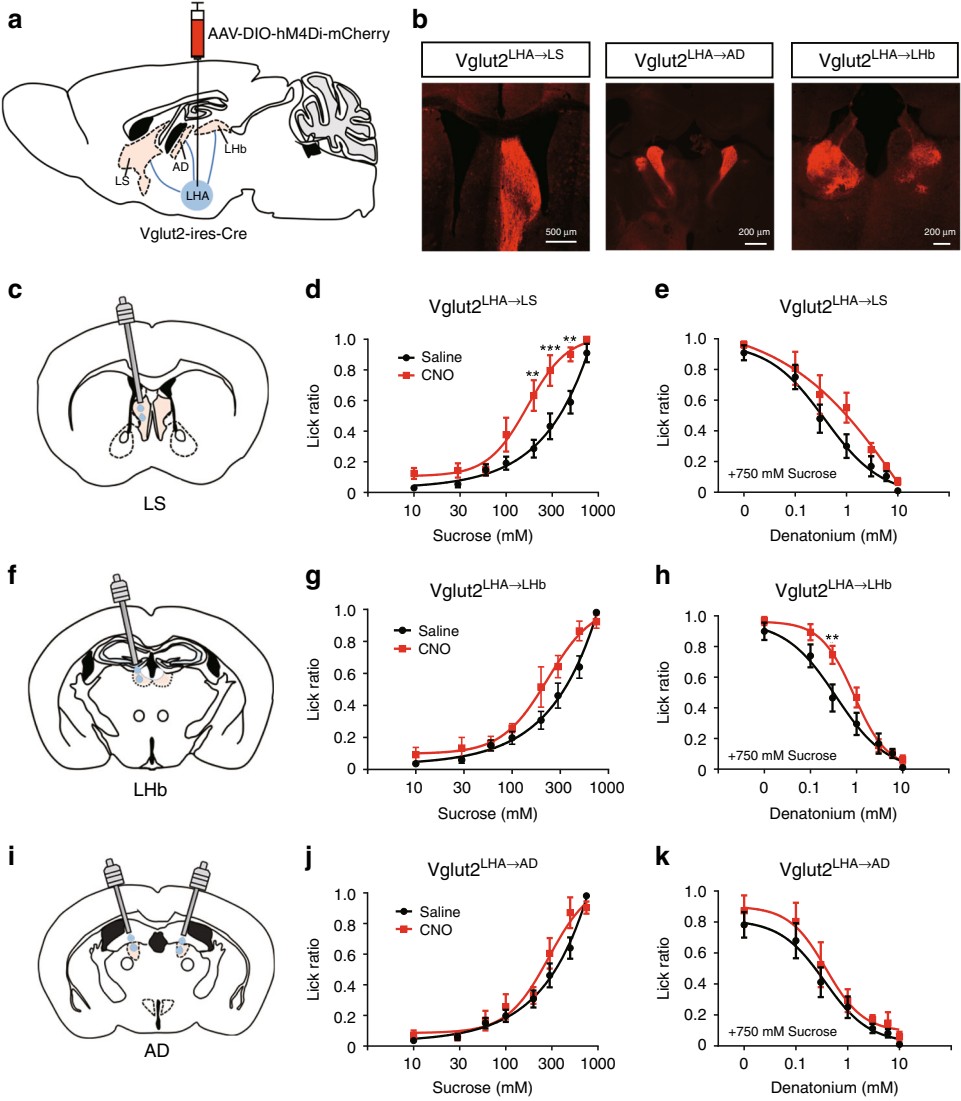

**Fig. 6** Two distinct hypothalamic pathways contribute to the modulation of sweet and bitter preferences. **a** Injection of AAV-DIO-hM4Di-mCherry into the LHA of Vglut2-ires-Cre mice. **b** Representative image of the projection regions (LS, AD, and LHb) of Vglut2$^{LHA}$ neurons. **c–k** Brief access taste tests after local inhibition of Vglut2$^{LHA}$ neurons projecting to the LS (**c**), LHb (**f**), or AD (**i**) by microinfusion of CNO, respectively. Preferences toward sweet taste (**d**, **g**, **j**) or bitter taste (**e**, **h**, **k**) were evaluated 10 min after microinjection of CNO (0.1 mg/ml, 200 nl). $n = 6$, $F = 34.32$, and $P = 8.9 \times 10^{-8}$ in **d**, $n = 6$, $F = 4.768$, and $P = 0.0326$ in **e**, $n = 6$, $F = 7.727$, and $P = 0.0071$ in **g**, $n = 6$, $F = 13.44$, and $P = 0.0005$ in **h**, $n = 5$, $F = 3.564$, and $P = 0.063$ in **j**, and $n = 5$, $F = 2.823$, and $P = 0.098$ in **k**, two-way ANOVA with Bonferroni post hoc test. All experiments were carried out with 10- to 16-week-old male mice. Data are given as means ± SEM. *$P < 0.05$, **$P < 0.01$, ***$P < 0.001$.

survive under hunger conditions. Mice show a stronger preference toward appetitive taste for efficient caloric food intake and are less sensitive to aversive taste for compromised eating.

Importantly, hunger-induced taste modification was recapitulated by a selective activation of LHA-projecting AgRP neurons (Fig. 2). Although taste modification has been investigated in the peripheral taste system in the context of hormonal effects[28], this is the first study to reveal a top–down gustatory regulation from the hypothalamus.

Sweet taste is composed of both sensory and hedonic aspects[21]. Our CTA experiments with optogenetics showed that LHA-projecting AgRP neurons do not affect sweet taste threshold. These results imply that these neurons modulate the hedonic aspect of sweet taste. Since palatability is defined as an indication of the hedonic value of food[29], LHA-projecting AgRP neurons might play a role in enhancing the palatability of sweet taste. As orofacial reaction reflects palatability even in mice[30], further study is required to clarify this point.

Among the various hypothalamic areas, the PVH is known to receive a dense innervation from AgRP neurons and to induce food intake (Supplementary Fig. 4B, I). Recent studies showed that the PVH plays an important role in fat–carbohydrate food choice behavior under hunger conditions[31]. Interestingly, our optogenetic mapping study shows that the LHA, but not the PVH, modifies taste preferences (Fig. 2f–k). Considering the fact that both PVH-projecting and LHA-projecting AgRP neurons are able to induce feeding behavior, these results indicate that multiple hypothalamic nuclei have individual roles in hunger to coordinately induce appropriate feeding behavior for survival.

A recent paper showed that the amygdala-projecting neurons in the gustatory cortex (GC) regulate taste-guided appetitive and aversive behaviors[32]. Our data did not show any change in taste preferences when activating the CEA terminals of AgRP neurons (Fig. 2l–n). These results imply that AgRP and GC neurons may innervate different populations of amygdala neurons.

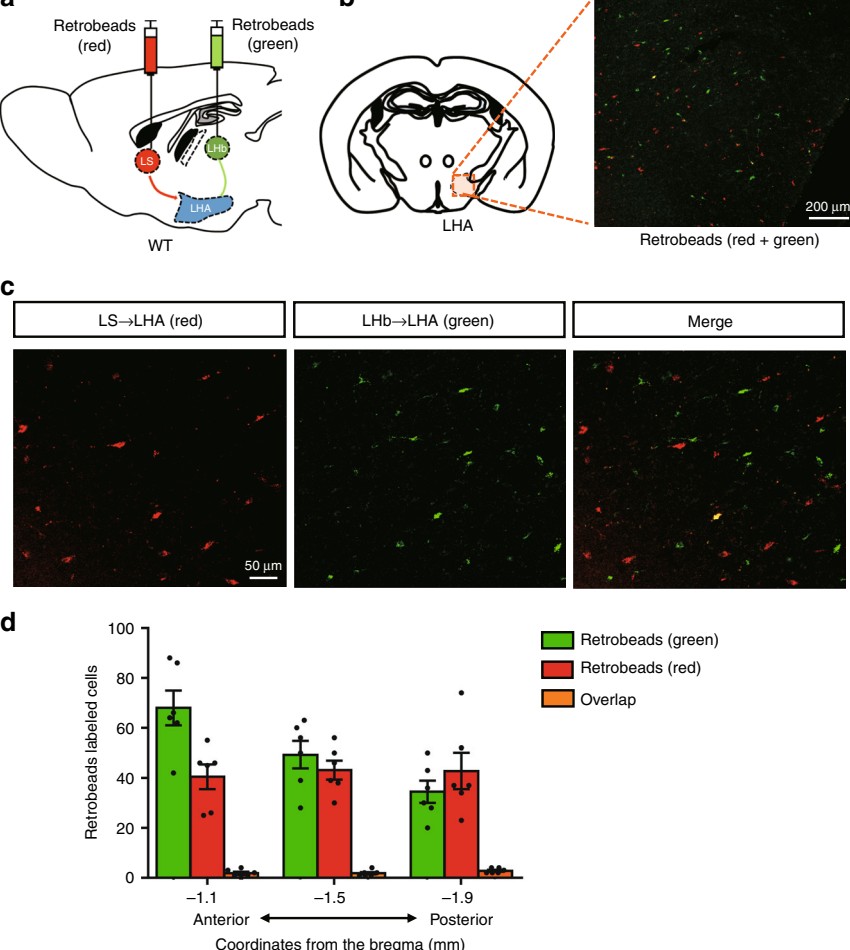

**Fig. 7** Different neuronal populations in the LHA project to the LS and LHb. **a** Schematic image of the retrobead injections into the LS (red) and LHb (green) in WT mice. **b** Retrograde transport of redbeads (from the LS) and green beads (from the LHb) in the LHA. **c** Representative enlarged confocal image of redbead-labeled cells (from the LS) and greenbead-labeled cells (from the LHb) in the LHA. **d** Quantification of red and green retrobead-labeled cells in the LHA along the anterior–posterior axis. Retrobead-positive cells were counted in three sections at different AP locations from $n = 2$ mice. Data are given as means ± SEM

The LHA is known to regulate feeding and motivational behaviors[33]. While there are a variety of neuronal cell types in the LHA, they are largely subdivided into glutamatergic Vglut2 neurons and GABAergic Vgat neurons. We show that Vglut2[LHA], but not GABAergic, neurons function as a downstream modulator of AgRP neurons to modify taste preferences (Fig. 5; Supplementary Fig. 7). Chemogenetic inhibition/activation studies indicated that these neurons bidirectionally modulate taste preference as well as appetite depending on the energy status.

Importantly, the pathway-selective chemogenetic inhibition study further revealed that two distinct subpopulations of Vglut2[LHA] neurons contribute to the modification of appetitive and aversive tastes (Fig. 6). While LS-projecting Vglut2[LHA] neurons selectively enhance sweet taste preference, LHb-projecting Vglut2[LHA] neurons regulate bitter taste sensitivity. These results are consistent with the electrophysiological results showing that the LHA contains appetitive taste-responsive neurons and aversive taste-responsive neurons[34]. In future studies, it will be meaningful to use the selective expression of the GCaMP6 calcium sensor into LS- or LHb-projecting Vglut2[LHA] neurons to monitor real-time neuronal activities toward various taste stimuli under fed or fasting conditions.

A genetic ablation study of AgRP neurons in adult mice indicated that loss of AgRP neurons led to an increase in *c-fos* expression in the LS and this increase was reversed by an infusion of a GABA agonist[22]. Importantly, the AgRP[ARC]→Vglut2[LHA]→LS neuronal circuit we demonstrated in this paper is consistent with these results. The LS is known to be a component of the limbic system and plays a modulatory role in reward by projecting to the ventral tegmental area[35]. While AgRP neurons are key components of homeostatic feeding, their projections to the LHA may play roles in enhancing sweet taste preference with positive emotion by interacting with the reward system via the LS.

The LHb is known to play an important role in regulation of the reward system by giving negative value signals[36–38]. The neural pathway from the LHA to the LHb is reported to be responsive to negative events contributing to goal-directed behaviors[39]. For example, aversive stimuli, such as foot shock, excite the LHb neurons and promote escape behaviors in mice. By contrast, silencing of the LHb neurons impairs aversion-induced escape behaviors. Since bitter taste is considered to be an aversive stimulus, this pathway is reasonable for the modulation of the taste aversion.

Retrograde-tracing experiments showed that distinct neuronal populations in the LHA project to the LS or the LHb, respectively (Fig. 7). Collectively, these results indicate the existence of two different types of lateral hypothalamic neuronal pathways for hunger-induced taste modification to bridge the gap between the homeostatic AgRP neuronal circuits and the reward system.

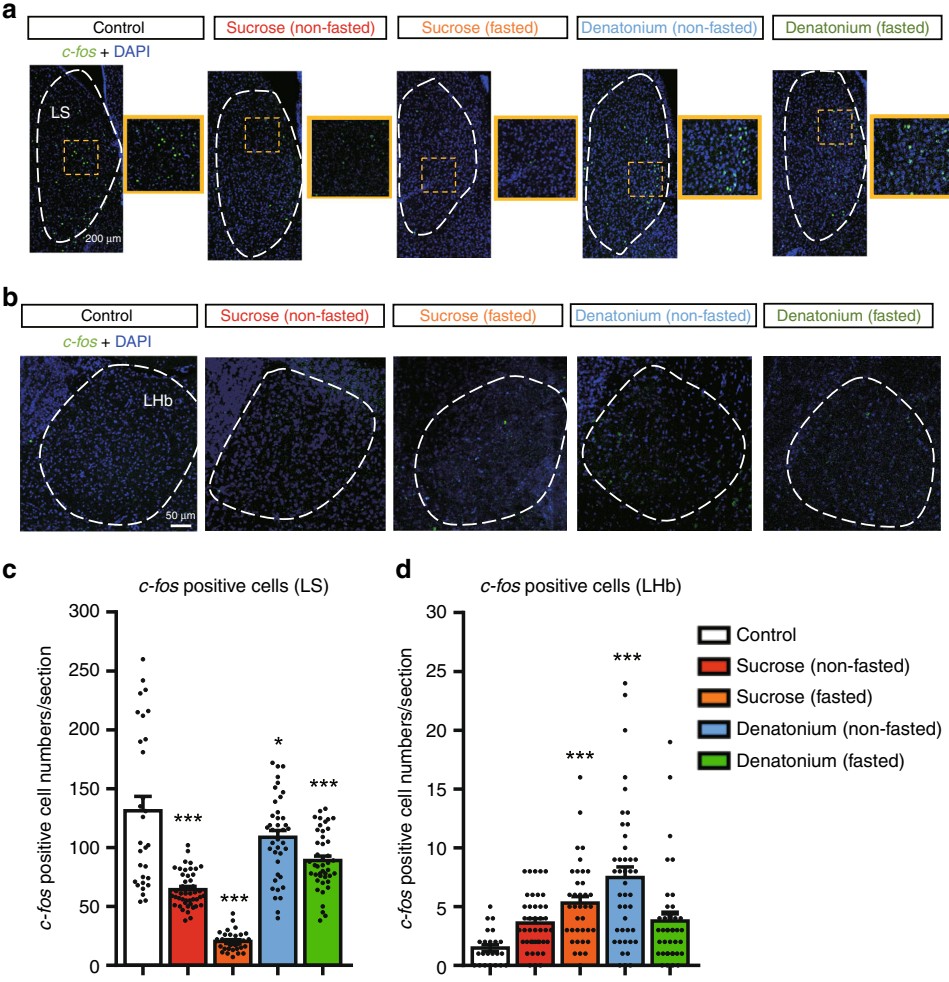

**Fig. 8** Fasting differentially affects taste-induced *c-fos* expression in the LS and the LHb. **a, b** Representative images showing *c-fos*-positive cells in the LS (**a**) and in the LHb (**b**). **c, d** Quantification of *c-fos*-positive cells in the LS (**c**) and in the LHb (**d**). The mice were treated with sucrose or bitter taste under fed or fasted conditions. $n = 3$–5 mice per group and ~10 brain slices per mouse were analyzed. $F = 51.25$ and $P = 4.9 \times 10^{-8}$ in **c**, $F = 11.1$ and $P = 0.0069$ in **d**, one-way ANOVA with Dunnett's post hoc test. All experiments were carried out with 10- to 16-week-old male mice. Data are given as means ± SEM. $*P < 0.05$, $**P < 0.01$, $***P < 0.001$ as compared with the control group

Sugars including sucrose contain sweet taste and nutrition. Recent evidence showed that sweet taste and its nutritional value cause dopamine release via the ventral and dorsal striatum, respectively, suggesting that they have different neuronal pathways in the reward system[40]. In this study, our results imply that the palatability of sucrose is strengthened by activation of the AgRP neurons. Thus, identification of the neuronal connections between the homeostatic AgRP neuronal circuits and the ventral striatum may be the next key point to fully understand how the brain modulates taste preference in a state-dependent manner.

Since the constitutive activity of AgRP neurons and change in taste preference are often observed in genetic mouse models of obesity and type 2 diabetes, such as db/db mice[41,42], the next step in this study is to evaluate whether the neuronal pathways we revealed here change under pathophysiological conditions. For instance, obese human patients were reported to have a strong preference for sweetness[43]. Reversing the biased food and taste preferences can be beneficial for human health.

## Methods

**Mice**. WT C57BL/6J mice were purchased from CLEA Japan. AgRP-ires-Cre mice (JAX-012899)[13,14,44–47] were purchased from Jackson Laboratory. Vglut2-ires-Cre mice were kindly provided by Dr. Bradford Lowell (Harvard University, JAX-016963). Vgat-ires-Cre mice were kindly provided by Dr. Yumiko Yoshimura

(National Institute for Physiological Sciences, JAX-016962). GAD2-Cre mice were kindly provided by Dr. Kazushige Touhara (the University of Tokyo). The animal experimental procedures followed the animal care guidelines approved by the University of Tokyo and by the National Institute for Physiological Sciences.

**Mouse maintenance and diet**. Mice were fed ad libitum and kept on a 12:12-h light:dark cycle. Unless stated otherwise, all experiments were carried out with male littermates that were 8–16 weeks old and maintained on a standard CE-2 mouse chow (CLEA Japan).

**Viruses**. AAV8-hSyn-DIO-hM3Dq-mCherry (Titer: $5.9 \times 10^{12}$ vg/ml), AAV1-FLEX-tdTomato (Titer: $5.0 \times 10^{12}$ vg/ml), and AAV8-hSyn-DIO-GFP (Titer: $5.2 \times 10^{12}$ vg/ml) were purchased from the University of North Carolina Vector Core. AAV8-hSyn-DIO-hM4Di-mCherry (Titer: $4.3 \times 10^{12}$ vg/ml) and AAV5-DIO-ChR2-EYFP (Titer: $7.7 \times 10^{12}$ vg/ml) were purchased from Addgene. AAV8-hEF1a-DIO-synaptophysin-mCherry (Titer: $2.48 \times 10^{12}$ vg/ml) was purchased from Virovek, Inc. AAV10-FLEX-rev-ChR2-tdTomato (Titer: $1.2 \times 10^{13}$ vg/ml) was purchased from the University of Pennsylvania School of Medicine.

**Viral injections**. Stereotaxic injections were performed as previously reported[45]. Briefly, mice were anesthetized with 1.5–2.0% isoflurane, and placed into a stereotaxic apparatus (David Kopf Instruments). The skull was exposed via a small incision, and a small hole was drilled (0.45-mm drill bit) into the skull for the injection of AAVs. A Hamilton 10-μl syringe with a 30-gauge blunt-end needle was inserted into the brain for virus delivery.

AAV-DIO-hM3Dq-mCherry, AAV-FLEX-ChR2-tdTomato, AAV-DIO-ChR2-EYFP, or AAV-hEF1a-DIO-synaptophysin-mCherry (300 nl per site) were bilaterally (or unilaterally in the case of AAV-hEF1a-DIO-synaptophysin-mCherry) injected

into the ARC (coordinates, bregma: AP: −1.46 mm, ML: ± 0.3 mm, and DV: −5.80 mm) of AgRP-ires-Cre mice at a speed of 50–100 nl/min by using a UMP3 pump regulated by Micro-4 (World Precision Instruments). AAV-DIO-hM3Dq-mCherry, AAV-DIO-hM4Di-mCherry, AAV-FLEX-tdTomato, or AAV-DIO-GFP (300 nl per site) were bilaterally injected (or unilaterally in the case of AAV-DIO-GFP) into the LHA (coordinates, bregma: AP: −1.34 mm, ML: ± 1.0 mm, and DV: −5.2 mm) of Vglut2-ires-Cre mice. Mice were allowed to recover for at least two weeks before starting food intake or brief access taste test experiments.

**Optic-fiber implantation**. For the optogenetic activation of AgRP$^{ARC}$ neurons, a single optic fiber (200-µm diameter, 0.22 NA, Thorlabs) was implanted slightly above the ARC (coordinates, bregma: AP: −1.46 mm, ML: ± 0.3 mm, and DV: −5.30 mm) and fixed in place by using dental cement (Densply). For the axon terminal stimulation study of AgRP neurons, an optic fiber was implanted over the PVH (coordinates, bregma: AP: −0.7 mm, ML: −0.15 mm, and DV: −4.6 mm), the LHA (coordinates, bregma: AP: −1.34 mm, ML: ± 1.0 mm, and DV: −4.7 mm), the CEA (coordinates, bregma: AP: −1.22 mm, ML: ± 2.5 mm, and DV: −4.35 mm), the paraventricular nucleus of the thalamus (PVT) (coordinates, bregma: AP: −1.1 mm, ML: 0 mm, and DV: −3.0 mm), or the bed nucleus of the stria terminalis (BNST) (coordinates, bregma: AP: + 0.62 mm, ML: ± 0.65 mm, and DV: −4.4 mm).

**In vivo photostimulation**. In vivo photostimulation was performed as previously reported[48]. Briefly, a 473-nm blue-light laser (COME2-LB474/300, Lucir) was used to deliver light pulses to the brain through fiber-optic cables (200-µm diameter, 0.22 NA, Doric lenses) firmly attached to implanted optic fibers. The power intensities of the laser at the optic fiber terminal were adjusted to approximately 10 mW/mm². For all of the photostimulation experiments, the following pulse protocol was used: 10-ms pulses, 20 pulses for 1 s, repeated every 4 s.

**Chemogenetic pathway-selective inactivation**. For pathway-selective inactivation by using hM4Di, a guide cannula (Plastics one) was implanted above each target: the LS (coordinates, bregma: AP: + 0.86 mm, ML: ± 0.92 mm, and DV: −3.57 mm at a 15° angle) unilaterally with a 4.5-mm guide cannula (Plastics one), the AD (coordinates, bregma: AP: −0.82 mm, ML: ± 1.49 mm, and DV: −1.85 mm at a 15° angle) bilaterally with a 2.6-mm guide cannula, and the LHb (coordinates, bregma: AP: −1.34 mm, ML: ± 1.03 mm, and DV: −1.72 mm at a 15° angle) unilaterally with a 2.6-mm guide cannula. Guide cannulas were firmly fixed to the skull by using an adhesive (Loctite). After a 3- to 4-week recovery period, 3 µM CNO (200 nL) was infused locally into the corresponding brain area at a speed of 300 nl/min through an internal cannula (Plastics one) connected to the guide cannula. CNO concentration was determined based on several reference papers[25,27,49]. The internal cannula was designed to target 0 mm below the tip of the guide cannula for the LS and 0.6 mm below the tip for the AD and the LHb.

**Food-intake measurements**. Mice were single housed for at least 1 week before food-intake measurements. For chemogenetic experiments in the light cycle, CNO (Sigma) was injected (1.0 mg/kg i.p.) between 9:00 AM and 10:00 AM (3–4 h after the beginning of the light cycle). After CNO treatment, mice were placed in new cages with 5–6 food pellets (2.5–4.0 g per pellet) of standard mouse chow. For chemogenetic experiments in the dark cycle, CNO was injected (1.0 mg/kg i.p.) 30 min before the beginning of the dark cycle. The mice were then placed in newly prepared clean cages with 5 or 6 food pellets (2.5–4.0 g per pellet) of standard chow. Food intake was measured 1 h and 2 h after CNO injection. For optogenetic experiments, food intake was measured at 0.5 h and 1 h from the start of photostimulation during the light cycle. CNO concentration was determined based on several reference papers[13,45].

**Brief access taste tests**. The brief access taste test was conducted in a plastic chamber with an opening on one wall for sipper tube access as previously reported[48,50]. Briefly, the stainless-steel sipper tube was approximately 4-cm long and was connected to a 15-ml plastic tube containing the taste solution. Water-deprived (23 h) mice were given 30-min training sessions per day for 3–5 days before testing began. During the recording of the licking behavior, the chamber bottom was covered with aluminum foil and the water spout was connected to an A/D converter (Intermedical, Japan) to record the times of the licks. Trained mice were deprived of water for 4–6 h before testing to evoke motivation for taste solution access. Number of licks were measured for 10 s from the first licking action of the sipper tube and performed with a series of concentrations for a certain taste solution as follows. For sweet taste, a sucrose (1–750 mM) or sucralose solution (0.08–25 mM) was provided in the order of decreasing concentration. For bitter taste, denatonium was dissolved in a 500 mM or 750 mM sucrose solution and provided in the order of increasing concentration (0.1–10 mM). For sour taste, citric acid was dissolved into a 500 mM sucrose solution and provided in the order of increasing concentrations (1–100 mM). The lick ratio was calculated relative to the results of the number of licks for a 500 or 750 mM sucrose solution per 10 s in each test session. Average maximum lick numbers were not significantly different in all the figures (WT mice (80 ± 2.4 licks, n = 7, Fig. 1b, c), AgRP-ires-Cre mice (72.9 ± 1.9 licks, n = 40, Fig. 1g, h, Fig. 2; Supplementary Fig. 1C; Supplementary

Fig. 2A-C; Supplementary Fig. 5, and Supplementary Fig. 6C, D, G, H), Vglut2-ires-Cre mice (79.6 ± 1.3 licks, n = 30, Figs. 4e, f, h, i and 5d, e, g, h, j, k), and Vgat-ires-Cre mice (67 ± 4.5 licks, n = 6, Supplementary Fig. 7D, E)). Dose–response curves were fitted with a nonlinear regression model by using Prism 6 software (GraphPad).

**Conditioned taste aversion (CTA) test**. The CTA test was performed with the methods described in Gaillard and Stratford[17] with a slight modification. Water-deprived (23 h) mice were trained for 30 min every day for 3–5 days before the CTA-learning session was initiated. During the CTA- learning session, water-deprived (23 h) mice were allowed to lick 300 mM sucrose during 8–10 10-s trials. Immediately after sucrose licking, the mice were i.p. injected with 225 mM LiCl to cause malaise (0.1 ml/10 g of body weight). We observed signs of malaise (lying flat on their bellies, moving slowly, coprophagia) in all the mice within 45 min after the LiCl injection. Mice were then given ad libitum access to water for 1 h, and then water deprivation was repeated until the next session. Acquisition of CTA was confirmed by the observation that the mice refused to lick the 300 mM sucrose solution. All the mice exhibited this phenotype after the 2- or 3-day CTA-learning session. To measure sweet taste sensitivity, a sucrose solution (3–500 mM) was presented in the order of increasing concentration toward water-deprived CTA mice. The lick ratio was calculated relative to the results of the number of licks for water in water-deprived (23 h) mice per 10 s in each test session.

**Slice electrophysiology and photostimulation**. Coronal slices of the hypothalamus (300-µm thick) were prepared from adult mice (aged 3–4 months) under deep anesthesia with isoflurane and kept in a normal artificial cerebrospinal fluid (ACSF) containing (mM): 126 NaCl, 3 KCl, 1.3 MgSO₄, 2.4 CaCl₂, 1.2 NaH₂PO₄, 26 NaHCO₃, and 10 glucose at 33 °C. AgRP neurons labeled with YFP in the ARC were targeted by patch pipettes under fluorescent and infrared differential-interference contrast optics (BX51, Olympus, Tokyo, Japan). The patch pipettes were filled with an internal solution containing (mM): 130 K-gluconate, 8 KCl, 1 MgCl₂, 0.6 EGTA, 10 HEPES, 3 MgATP, 0.5 Na₂GTP, and 10 Na-phosphocreatine (pH 7.3 with KOH). Membrane potentials were recorded in the current-clamp mode by using a Multiclamp 700B amplifier (Molecular Devices, CA, USA). To conduct an analysis of the membrane potentials, tetrodotoxin (1 µM) was added to block the action potentials. We selected cells with a high seal resistance (>1 GΩ) and a low series resistance <35 MΩ. For the extracellular recordings, action potentials were recorded from the soma by using patch pipettes in the loose-seal cell-attached mode.

To activate ChR2, blue-light flashes (440 nm, using a diode laser) were delivered through an air objective (×4, 0.16 NA) to slices. The diameter of the light beams was ~20 µm. The photostimuli consisted of 5 light pulses of 10-ms durations at 20 Hz at a power of 0.3 mW on the surface of slices. The light pulses were applied to each of 7 × 7 sites (50-µm apart) surrounding the soma at intervals of 3 s in a quasi-random sequence. The photostimulation was repeated 3–5 times. To assess the firing rate during the photostimulation, we measured the number of spikes in the 0–400-ms time window after the first pulse for each stimulation site and selected the stimulation sites by using the criteria that the photostimulation induced the top 50% of the changes in the firing rate among the 49 stimulation sites.

**Taste stimuli and _c-fos_ experiments**. Taste stimulation experiments were performed as previously reported[48]. Briefly, mice were acclimated to handling by an experimenter for four consecutive days (5 min/day) before starting the experiments. Half of the animals were overnight fasted before taste stimulation. The experimenter provided each mouse with 1 ml of sweet (500 mM sucrose) or bitter (1 mM denatonium) solution by dividing into several times. Approximately 1 h after taste stimulation, the animals were perfused to perform immunostaining for _c-fos_.

**Brain tissue preparation**. Mice were deeply anesthetized with 5.0% isoflurane and perfused with 4% paraformaldehyde in 0.1 M phosphate-buffered saline (PBS, pH 7.4). Brain tissue was postfixed in this solution overnight and transferred to 30% sucrose in PBS. Brains were sectioned on a microtome (REM-700, YAMATO KOHKI) at 50 µm. Brain slices were stored in cryoprotectant solution (30% sucrose (w/v), 30% ethylene glycol (v/v), 1% PVP-40 (w/v), and 50 mM PBS) at −25 ºC.

**Immunohistochemistry**. After washing three times in PBS containing 0.1% Triton X-100 (PBST), brain slices were incubated overnight at 4 ºC with primary antibodies diluted in PBST containing 5% normal donkey serum. Slices were then washed three times and incubated with fluorophore-conjugated secondary antibodies for 2 h at room temperature. Slices were rinsed three times in PBST and then mounted with Vectashield (Vector Labs). Fluorescence images were taken with VS120 slide scanner or FV3000 confocal microscope (Olympus).

**Antibodies**. For primary antibodies, Goat anti-AgRP (1:1000, Neuromics), rabbit anti-AgRP (1:1000, Pheonix pharmaceuticals), goat anti-c-Fos (1:400, Santa Cruz),

and rat anti-RFP (1:1000, ChromoTek) were used. For fluorophore-conjugated secondary antibodies, donkey anti-goat Alexa fluor 488 (1:300, Life Tech), donkey anti-rabbit Alexa fluor 488 (1:300, Life Tech), and donkey anti-Rat Alexa fluor 568 (1:300, Abcam) were used.

**Monosynaptic retrograde rabies tracing**. AAV serotype 2 CAG-FLEx-TCB (for TVA-mCherryBright) ($1.2 \times 10^{13}$ gp/ml) and AAV serotype 2 CAG-FLEx-RG ($2.4 \times 10^{12}$ gp/ml) were generated de novo by using the plasmid described previously[51] by the UNC vector core.

Preparation of rabies virus was conducted by using the RVΔG-GFP and B7GG and BHK-EnvA cells[52]. The EnvA-pseudotyped RVΔG-GFP + EnvA titer was estimated to be $1.47 \times 10^{10}$ infectious particles/ml based on serial dilutions of the virus stock followed by infection of the HEK293-TVA800 cell line (a gift from Dr. Callaway).

For trans-synaptic tracing by using the rabies virus, about 150 nL of a 1:3 mixture of AAV2 CAG-FLEx-TCB and AAV2 CAG-FLEx-RG was injected into the LHA of GAD2-Cre and Vglut2-ires-Cre mice (coordinates, bregma: AP: −1.0 mm, ML: ± 0.8 mm, and DV: −5.0 mm). Three weeks later, 50 nL of rabies ΔG-GFP + EnvA (glycoprotein-deleted rabies expressing GFP) was injected into the same brain region to initiate trans-synaptic tracing. Seven days later after the rabies injection, mice were processed for immunohistochemistry.

**Retrobeads**. Retrograde neuronal labeling was performed by injecting 150 nl of red or green retrobeads (Lumafluor) at the coordinates of the projection areas of the Vglut2$^{LHA}$ neurons (the LS or the LHb). After allowing 7 days for retrograde transport, animals were processed for immunohistochemistry.

**Chorda tympani (CT) nerve recordings**. A whole gustatory nerve response from the CT nerve was obtained as described previously[53]. Briefly, mice were anesthetized by using sodium pentobarbital and urethane. Either saline or CNO (1.0 mg/kg i.p.) was injected 10 min before the anesthetic procedure. A tracheal cannula was implanted in each animal, and the animal was secured with a headholder. The CT nerve was exposed at its exit from the lingual nerve by removal of the internal pterygoid muscle, dissected free from the surrounding tissues, and cut at the point of its entry into the bulla. The entire nerve was placed on a platinum wire electrode. An indifferent electrode was positioned nearby in the wound. Whole-nerve activities were amplified, displayed on an oscilloscope, and monitored with an amplifier (DAM50; World Precision Instruments Inc., Sarasota, FL). The amplified signal was passed through an integrator with a time constant of 1 s. The magnitude of the whole-nerve response was measured as the height of the integrated response from baseline (before stimulation) at approximately 5 s after the onset of stimulation to avoid any tactile effect of the stimuli. The taste solution was applied for 30 s, followed by a >30-s rinse with deionized water. The CT nerve response to taste solutions was recorded approximately 90–180 min after saline or CNO injection. Application of each taste solution was repeated at least twice, and the mean response was calculated. Tastant solutions for the CT nerve response were (in mM): 30–500 sucrose, 3–30 sucralose, and 3–30 denatonium benzoate (denatonium). The relative response magnitude for each tastant was calculated relative to 100 mM ammonium chloride as a control. All experiments were carried out with 10- to 20-week-old male AgRP-hM3Dq mice ($n = 5–6$ per group).

**Cell count**. In the retrobead-labeling analysis, cells were counted manually. For each coordinate (bregma −1.1 mm, −1.5 mm, and −1.9 mm), three slices were used ($n = 2$ mice, Fig. 6). In taste-induced c-fos analysis (Fig. 8c, d), c-fos-positive cells were counted manually from control ($n = 3$), sucrose (non-fasted) ($n = 3$), sucrose (fasted) ($n = 5$), denatonium (non-fasted) ($n = 3$), and denatonium- (fasted) ($n = 3$) treated mice. For each mouse, approximately 10 slices for the LS or LHb were used for analysis.

**Schematic images**. The schematic brain images were modified from a mouse brain atlas[54].

**Statistical analysis**. Statistical analysis was performed with Prism 6.0 software (GraphPad). Differences across more than two groups were analyzed with one-way analyses of variance (ANOVA) followed by Dunnett's post hoc test (Fig. 8c) or two-way ANOVA followed by a Bonferroni post hoc test (Figs. 1b, c, g, h, j, k, 2, 3d, 5e, f, h, i, and 6; Supplementary Fig. 1, Supplementary Fig. 2, Supplementary Fig. 3, Supplementary Fig. 5, Supplementary Fig. 6, Supplementary 7D, E) for multiple comparisons as described in the figure legends. For comparison between two groups, two-tailed paired (Figs. 1f, i, 3c and 5d, g; Supplementary Figs. 4 and 7C) Student's $t$ tests were performed as described in the figure legends. Data are expressed as means ± standard error of the mean (SEM) for the indicated number of observations. We defined $P < 0.05$ as statistically significant.

**Reporting summary**. Further information on research design is available in the Nature Research Reporting Summary linked to this article.

## Data availability

The data that support the findings of this study are available from the corresponding author on reasonable request.

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

## Acknowledgements

We thank R. Kogure, T. Fukuda, K. Fujiwara, and A. Takahashi for the maintenance of the mouse colonies. This research was supported in part by Grants-in-Aid for Scientific Research (15H05624 and 18H02160 to K.N.; 18H05267 to K.T.) from the Ministry of Education, Culture, Sports, Science, and Technology of Japan, by the Cross-Ministerial Strategic Innovation Promotion Program (SIP) (14532924 to K.N. and T.M.), by the Lotte Shigemitsu Prize (to K.N.), and by ERATO Touhara Chemosensory Signal Project from JST to K.T. (JPMJER1202).

## Author contributions

O.F., T.M., Y.M., and K.N. conceived and designed the study. O.F. carried out AAV or retrobead injections, mouse behavioral experiments together with K.N., and immuno-histochemical analyses with help from Y.I. M.N. performed the gustatory nerve recording. A.W.I. performed the slice electrophysiolgical studies. K.K.I. and K.M. carried out rabies injections. Y.Y. and K.T. provided key reagents and critical advice. O.F. and K.N. wrote and revised the paper, and all of the authors edited the final draft.

## Competing interests

The authors declare no competing interests.
