## [Peer Review File · Nature Communications]

Reviewers' Comments:

Reviewer #1:

Remarks to the Author:

It appears that the central neural circuits for taste modifications in response to fasting are not well understood. The paper proposes novel hypothalamic pathways, including cell populations in the arcuate nucleus and lateral hypothalamic area, projecting to the lateral septum or habenula that mediate appetitive or aversive taste sensitivities, respectively. The paper utilizes chemogenetic and optogenetic methods in combination with tracing techniques to study these neural circuits. I think that the paper is potentially important.

The following comments may help to strengthen the conclusions:

a) It appears that the AgRP-ires-Cre mice are a commercial strain. The authors do not provide any reference or characterization of this strain. For example, it would be helpful to check that expression of DREADD and ChR2 is specific for AgRP neurons by doing double immunostaining for the peptide and the optical or chemical system.

b) The study does not reveal in-vivo activities of the hypothalamic circuits tracking appetitive or aversive taste sensitivities in hungry and satiated mice. Therefore, chemogenetic or optogenetic activation of AgRP neurons may be artificial. I think it would be meaningful to also chemogenetically inhibit these neurons in fasted mice.

The authors may also attempt to study the activity of their circuits under physiological conditions. Ideally, this is done by using in-vivo electrophysiology or imaging. At least, the authors can use classical c-fos immunohistochemistry to show if activity (or the lack thereof) of the distinct sub-populations of LS- or Lhb-projecting LHA neurons correlates to changes in appetitive and aversive taste sensitivities in response to fasting.

c) The LHA also contains, besides glutamatergic neurons, other cell populations, including GABAergic neurons. As a matter of fact, the authors state that "the LHA-projecting AgRP neurons connect to a part of Vglut2LHA neurons" and figures 3 and S7 indicate that other cell populations than the Vglut2 neurons may be involved in their circuits. The statement that "inhibition of vGAT neurons in the LHA led to a decrease in food take" is not a very satisfying reason to focus solely on Vglut2 neurons. In this respect, I also think the statement that Vglut2 LHA "neurons are sufficient and necessary to modify sensitivities for both appetitive and aversive tastes" is overstated. In fact, although the authors do not indicate in figure 4F whether the difference between fasted mice after chemogenetic activation of glutamatergic LHA neurons and fed mice is significant or not, activation of the glutamate neurons does not appear to normalize the aversive taste behavior of hungry mice. I feel that it would be meaningful to investigate the role of GABAergic LHA neurons in regulating appetitive or aversive taste.

d) Do the authors have in-vitro electrophysiology (patch) data to show that the M3 or ChR2 transduced AgRP cells respond to CNO or light, respectively? The c-Fos data are compelling but of course, could reflect only changes in Ca²⁺_i, not increased firing. Electrophysiology (patch) seems to be a relatively common validation when using chemogenetic and optogenetic approaches. Especially, IPSC effects in LS- and Lhb-projecting LHA neurons after terminal stimulation of AgRP neurons could be examined.

e) Why do the authors increase the sucrose content (from 500 mM to 750 mM) in the aversive taste experiments from figure 4 onward? Is antidromic activation of AgRP neurons after terminal stimulation, inefficient chemogenetic inhibition of LHA Vglut2 neurons, or the contribution of other LHA neurons the reason for the difference in aversive taste sensitivity?

f) Figure 3A and B are redundant. Unless there is a point in showing coronal and sagittal sections, either the coronal or sagittal panels can be removed. Instead it would be informative to indicate Vglut2 neurons that receive inputs from AgRP neurons and eventually, count neurons with or without AgRP input.

g) I am not convinced that the data shown in figure 5E are not significant; in other words, LS-projecting LHA neurons may modulate bitter taste in response to hunger. In addition to the unpaired Student's t-test, two-way ANOVA should be performed on the behavioral data.

h) The authors may provide histological evidence for accurate cannula placement or injection for the micro-infusion (figure 5) and retrobeads (figure 6) experiments, respectively.

i) Micro-infusion of CNO occurs near ventricles and thus, it is possible that CNO leaks into the ventricle and reaches other brain areas. Moreover, there is no evidence that this technique works in the AD. c-fos in LS, LHb and AD neurons may be sufficient to show specific postsynaptic activation by glutamatergic LHA neurons.

Minor comments:

a) The authors might consider adding a reference or two to justify the dose of CNO used.

b) The paragraph about leptin in the introduction seems to be out of context. Is it necessary?

c) I think the labeling 'LS' in figure 5F is wrong.

Reviewer #2:

Remarks to the Author:

The manuscript from Fu et al investigates the neural circuits involved in hunger-related changes in the consumption of palatable and aversive stimuli. The experiments rely on an array of techniques ranging from behavioral analysis, optogenetics, chemogenetics, anatomical labeling, and viral strategies in transgenic mice. The results demonstrate that: 1) hunger increased licking for intermediated concentrations of sweet and bitter stimuli; 2) activation of AgRP-expression neurons in the arcuate nucleus of the hypothalamus recapitulates the effects of hunger; 3) specifically the effect is mediated by AgRP neurons projecting to glutamatergic neurons in the lateral hypothalamus (LHA); 4) LHA projection neurons mediate the effects of hunger on the consumption of sweets and bitters by targeting the lateral septum (LS) and the lateral habenula (LHb) respectively.

The results presented in this manuscript are interesting and very relevant as they unveil novel mechanisms responsible for modulating consumption of palatable and aversive tastants in different physiological states. The experiments are rigorously conducted and well controlled. The manuscript is clearly written and follows a linear logic. I do however have some concerns and comments on data analysis and interpretation that need to be addressed.

- The results are framed as evidence for the mechanisms of hunger-induced taste modifications. However, the authors do not demonstrate that hunger affects taste sensitivity. The results of the brief access test could equally be due to changes in the perceived palatability of stimuli. The brief access test does not dissociate sensory and hedonic components. If the authors want to directly address taste sensitivity, they have to perform a taste detection task (e.g., a 2 alternative forced choice task where the animal has to discriminate different concentrations of tastants versus water). I do not think these

experiments are necessary, but I do think that the authors should avoid attributing their results to changes in sensory sensitivity (hence they should edit the text accordingly). I also encourage the authors to add a discussion on the contribution of sensory (i.e., gustatory) and hedonics effects to their results.

- Related to the point above, do the authors have any orofacial reactions data? If so, they could provide information on hunger-mediated changes in palatability. If they do not have these data, they should at least discuss the role of palatability in the discussion.

- References to the literature on hunger-mediated effects on taste processing are very limited. Fundamental papers, like de Araujo et al Neuron 2006, are not cited. The authors should do a more thorough job with their citations.

- The results in Figure 3 showing axons of LHA-projecting AgRP neurons "contacting" Vglut2positive neurons should be rigorously quantified. In the absence of a clear quantification of puncta, this evidence should be interpreted more cautiously.

- Statistical analyses of data from brief access tests rely on t-tests. The authors do not mention if the tests were corrected for multiple comparisons. If they did not use this correction, they should include it in the revision.

- Figure 5F: the label should be LHb and not LS

Reviewer #3:

Remarks to the Author:

In this study, the authors set out to examine the neural substrates responsible for hunger induced taste modification. The authors demonstrate that activation of the AgRP neurons increases licking for sugar water of moderate concentration and increases the tolerance for bitter. They further show the similar behavioral changes during activation of AgRP to LH projections and direction inhibition of LH glutamatergic cells. Finally, they show that inactivation the LH projection to LS increases licking for sugar while inactivating the LH projection to LHb increases tolerance for the bitter.

The major conceptual issue is that the reviewer finds the results do not support a role of the studied cells/pathways in modulating the sensitivity to sugar taste, instead it supports a role of the cells/pathways in increasing motivation to obtain sugar. In reviewer's understanding, changes in sensitivity to the sugar taste means that the threshold to detect the sugar decreases. For example, fed animal cannot tell difference between 10mM sucrose solution from water while the starved animal can. In this study, the main difference between Agrp cell activated animals and control animals are their licking rate for moderately concentrated sugar water (e.g. 100mM). This concentration is apparently well above the detecting threshold for sugar as control animals showed clear increase in licking at that concentration in comparison to water. Thus, the increase in licking supports an increase in motivation for sugar water but not an increase in sensitivity to sugar taste. If the sensitivity to sugar is indeed increased, the reviewer will expect an increase in licking sugar water of low concentration. On the other hand, the reviewer found the claim that AgRP activation decreases bitter sensitivity is more convincing given the data in Supplementary Figure 2B. In this experiment, it does appear that AgRP activation decreased the detection threshold for the bitter as the control water-restricted animal reduced licking of water containing 0.1mM Denatonium while the AgRP activated animals did not. Also, since AgRP neurons do not drive drinking (but the cells do drive feeding and thus confound the interpretation for increased licking of sucrose water), it excludes the possibility that

the increased licking is simply due to an increase in thirst.

Other issues include:

1. The claim that the AgRP neurons target LH glutamatergic cells need to be shown more convincingly using slice recording or monosynaptic rabies tracing. Do AgRP cells target GABAergic cells in the LH or they only target glutamatergic cells?
2. The paper will be significantly strengthened by showing an increased response of the AgRP cells and LH glutamatergic cells to sugar water at borderline concentration. Based on the hypothesis, one will predict that LH glutamatergic cells will not respond to low concentration sugar water in fed animal but will respond to it in fasted animals or AgRP activated animals.
3. As CNO is a small molecular that is readily diffused throughout the body, local injection of CNO may affect adjacent regions. It will be better to use an intersectional retrograde labeling strategy to target the LH glutamatergic -LHb projecting and LS projecting cells especially given that they have low overlap. It will be even more informative to record the activity of LH glutamatergic -LHb projecting and LS projecting cells to understand whether they show differential responses to sugar and bitter taste.

Response to the Referees' comments

Nature Communications manuscript: NCOMMS-18-24432

TITLE: Hypothalamic neuronal circuits regulating hunger-induced taste modification
by O. Fu *et al.*

We would like to thank the three reviewers for their constructive criticisms, which have helped to further improve the quality of the manuscript.

=====

Response to the points raised by Reviewer #1

Reviewer #1 (Remarks to the Author):

It appears that the central neural circuits for taste modifications in response to fasting are not well understood. The paper proposes novel hypothalamic pathways, including cell populations in the arcuate nucleus and lateral hypothalamic area, projecting to the lateral septum or habenula that mediate appetitive or aversive taste sensitivities, respectively. The paper utilizes chemogenetic and optogenetic methods in combination with tracing techniques to study these neural circuits. I think that the paper is potentially important.

The following comments may help to strengthen the conclusions:

1) REVIEWER'S COMMENT: It appears that the AgRP-ires-Cre mice are a commercial strain. The authors do not provide any reference or characterization of this strain. For example, it would be helpful to check that expression of DREADD and Chr2 is specific for AgRP neurons by doing double immunostaining for the peptide and the optical or chemical system.

OUR RESPONSE: According to the suggestion, we cited papers from several groups including us, (Tong *et al* 2008, Krashes *et al* 2011, Betley *et al* 2013, Chen *et al* 2015, Nakajima *et al* 2016, Reichenbach *et al* 2018) because they used the same AgRP-ires-Cre knockin mouse line (JAX 012899). As one example, here we briefly explained the characterization of the AgRP-ires-Cre mouse line by Reichenbach *et al*. Since AgRP neurons co-express neuropeptide Y (NPY), Reichenbach *et al* crossed the AgRP-ires-Cre mice with the NPY-hrGFP transgenic mice. After that, the AgRP-ires-cre::NPY GFP mice were further crossed with the rosa26-td-Tomato reporter

mice. It revealed that more than 90% of NPY neurons in the ARC express tdTomato as a marker of Cre-dependent recombination in AgRP-expressing neurons. Thus, we believe that this mouse line selectively expresses Cre in AgRP neurons.

These additional references have been written in the Methods section (page 15 line 3) and added into the reference section of the revised manuscript.

2) REVIEWER'S COMMENT: The study does not reveal in-vivo activities of the hypothalamic circuits tracking appetitive or aversive taste sensitivities in hungry and satiated mice. Therefore, chemogenetic or optogenetic activation of AgRP neurons may be artificial. I think it would be meaningful to also chemogenetically inhibit these neurons in fasted mice. The authors may also attempt to study the activity of their circuits under physiological conditions. Ideally, this is done by using in-vivo electrophysiology or imaging. At least, the authors can use classical *c-fos* immunohistochemistry to show if activity (or the lack thereof) of the distinct sub-populations of LS- or LHb-projecting LHA neurons correlates to changes in appetitive and aversive taste sensitivities in response to fasting.

OUR RESPONSE: To evaluate whether inhibition of AgRP neurons lead to compete with hunger-induced taste modification, we chemogenetically suppressed AgRP neurons under physiological hunger condition. We found that activation of the inhibitory DREADD in AgRP neurons reverses the hunger-induced taste change (Fig. 1J and 1K).

To investigate the activity of our identified neuronal circuits under physiological conditions, we performed immunohistochemical analysis of *c-fos* in the LS and the LHb from the fed or fasted mice treated with sweet or bitter taste. While hunger and sweet taste cooperatively suppressed basal *c-fos* expression in the LS, bitter taste did not show such effects in the LS (Fig. 8A). By contrast, hunger selectively suppressed bitter induced *c-fos* expression in the LHb. Such suppression was not observed in the case of sweet taste in the LHb (Fig. 8C). These results suggest that hunger differentially modulates taste induced *c-fos* expression in the LS and in the LHb.

These new data have been incorporated into the Results section of the revised manuscript (Pages 5-6 lines 28-2: "We next examined whether suppression of AgRP neurons affects taste preference...") and (Page 11 line 10 "Fasting differentially modulates taste induced *c-fos* expression in the LS and the LHb").

Moreover, new figures have been added (Fig. 1E, I-K and Fig. 8). The procedure that we used is described under the Methods section on page 19 line 17 (“Taste stimuli and *c-fos* experiments”).

3) REVIEWER’S COMMENT: The LHA also contains, besides glutamatergic neurons, other cell populations, including GABAergic neurons. As a matter of fact, the authors state that “the LHA-projecting AgRP neurons connect to a part of Vglut2LHA neurons” and figures 3 and S7 indicate that other cell populations than the Vglut2 neurons may be involved in their circuits. The statement that “inhibition of vGAT neurons in the LHA led to a decrease in food take” is not a very satisfying reason to focus solely on Vglut2 neurons. In this respect, I also think the statement that Vglut2 LHA “neurons are sufficient and necessary to modify sensitivities for both appetitive and aversive tastes” is overstated. In fact, although the authors do not indicate in figure 4F whether the difference between fasted mice after chemogenetic activation of glutamatergic LHA neurons and fed mice is significant or not, activation of the glutamate neurons does not appear to normalize the aversive taste behavior of hungry mice. I feel that it would be meaningful to investigate the role of GABAergic LHA neurons in regulating appetitive or aversive taste.

OUR RESPONSE: To directly evaluate the role of GABAergic LHA neurons, we carried out a series of taste guided licking experiments with *Vgat-ires-Cre* mice that selectively express Cre recombinase in GABAergic neurons (Supplemental Fig. 7). Analogously to the case of *Vglut2^{LHA}* neurons, we introduced the inhibitory DREADD into GABAergic LHA neurons and then brief access taste tests were performed. Importantly, inhibition of GABAergic neurons did not affect any taste preference (Supplemental Fig. 7D and 7E). These results suggest that *Vglut2*, but not GABAergic, LHA neurons function as a downstream regulator of AgRP neurons in hunger-induced taste modification.

These new data have been incorporated into the Results section of the revised manuscript (page 9 line 25, GABAergic neurons in the LHA are not required for the modulation of taste preferences). Moreover, new figures have been added (Supplemental Fig. 7 in the revised version).

In Fig. 5I in the revised manuscript, while Fasted (CNO) and Fed (saline) groups exhibited very lower lick ratio scores compared to that of Fasted (saline) group, statistical analysis indicated that there is a significant difference ($P=0.033$) between

Fasted (CNO) and Fed (saline) groups. We have thus changed several sentences in the revised manuscripts as follows:

Abstract, page 3 lines 1-2, and page 9 lines 22-23: “Vglut2^{LHA} neurons play a key role in...”

Page 9 lines 20-21: “Vglut2^{LHA}-hM3Dq mice treated with CNO largely lost....”

4) REVIEWER’S COMMENT: Do the authors have in-vitro electrophysiology (patch) data to show that the M3 or Chr2 transduced AgRP cells respond to CNO or light, respectively? The c-Fos data are compelling but of course, could reflect only changes in Ca²⁺i, not increased firing. Electrophysiology (patch) seems to be a relatively common validation when using chemogenetic and optogenetic approaches. Especially, IPSC effects in LS- and LHb-projecting LHA neurons after terminal stimulation of AgRP neurons could be examined.

OUR RESPONSE: To evaluate the functional expression of Chr2 in AgRP neurons *in vitro*, we recorded action potentials from the Chr2-expressing AgRP neurons in acute hypothalamic slices (Supplemental Fig. 4). The firing rate for the recorded neurons was significantly increased during photostimulation compared with that before the stimulation. In addition, we recorded the membrane potentials from Chr2-expressing AgRP neurons with a whole cell recording in the current clamp mode in the presence of TTX. The membrane potentials were depolarized during the photostimulation. The peak of the depolarization evoked by the first pulse of photostimulation at the most effective stimulation sites was significantly higher than the resting membrane potential before photostimulation. The photostimulation did not affect the firing rate or membrane potentials of the Chr2-YFP-negative neurons in the ARC. These results showed that whole-cell patch clamp electrophysiological recordings with brain sections containing Chr2-expressing AgRP neurons showed that Chr2-expressing AgRP neurons were efficiently activated with light in our preparation.

These new data have been incorporated into the Results section of the revised manuscript (pages 6-7 lines 25-3). Moreover, new figures have been added (Supplemental Fig. 4D and 4E in the revised manuscript). The procedure that we used is described under the Methods section (page 18 line 27: “Slice electrophysiology and photostimulation”).

5) REVIEWER’S COMMENT: Why do the authors increase the sucrose content (from 500 mM to 750 mM) in the aversive taste experiments from figure 4 onward? Is

antidromic activation of AgRP neurons after terminal stimulation, inefficient chemogenetic inhibition of LHA Vglut2 neurons, or the contribution of other LHA neurons the reason for the difference in aversive taste sensitivity?

OUR RESPONSE: In Vglut2-ires-Cre mice, we increased the sucrose concentration in both appetitive and aversive experiments (See Fig. 5 EFI in the revised manuscript). The reason for this is that we would like to confirm the saturated lick response in the dose-dependent increase in sweet taste (as shown in Fig. 5E). For this purpose, a high concentration of sucrose solution (750 mM) is required for Vglut2-ires-Cre mice. This may be due to difference in mouse strain (AgRP-ires-Cre versus Vglut2-ires-Cre). As the reviewer pointed out, chemogenetic inhibition using hM4Di is unlikely to be 100% perfect. As shown in Fig. 1I, the inhibitory effects of hM4Di are often partial in behavioral experiments (See Ref. 13 Krashes *et al* 2011 JCI paper Fig. 2G as another example). Thus, incomplete inhibition of Vglut2^{LHA} neurons may be an alternative possibility. Based on the results of the role of GABAergic LHA neurons in taste modification (Supplemental Fig. 7), we do not think that non-Vglut2 neurons caused this change.

6) REVIEWER'S COMMENT: Figure 3A and B are redundant. Unless there is a point in showing coronal and sagittal sections, either the coronal or sagittal panels can be removed. Instead it would be informative to indicate Vglut2 neurons that receive inputs from AgRP neurons and eventually, count neurons with or without AgRP input.

OUR RESPONSE: According to this suggestion, we have removed Figure 3B in the revised manuscript. To further confirm direct synaptic connections between AgRP neurons and Vglut2^{LHA} neurons, we carried out monosynaptic rabies tracing with Vglut2-ires-Cre mice. Retrograde transsynaptic labeling from Vglut2^{LHA} neurons showed direct synaptic connections between AgRP neurons and Vglut2^{LHA} neurons as shown in Fig. 4C.

These new data have been incorporated into the Results section of the revised manuscript (pages 8-9 lines 27-1: "To further confirm the connection between AgRP neurons and Vglut2^{LHA} neurons, we next carried out monosynaptic rabies tracing experiments..."). Moreover, new figures have been added (Fig. 4B and 4C in the revised manuscript). The procedure that we used is described in the Methods section (page 20 line 14: "Monosynaptic retrograde rabies tracing").

7) REVIEWER'S COMMENT: I am not convinced that the data shown in figure 5E are not significant; in other words, LS-projecting LHA neurons may modulate bitter taste in response to hunger. In addition to the unpaired Student's t-test, two-way ANOVA should be performed on the behavioral data.

OUR RESPONSE: According to the suggestion, we performed more appropriate statistical analyses of the whole data in the revised manuscripts. As a result, statistical analyses with two-way ANOVA and *post hoc* Bonferroni test showed no significant differences in bitter response by inhibition of LS-projecting LHA neurons in Fig. 5E in the original manuscript (Fig. 6E in the revised manuscript). The statistical analysis that we used is described in the Methods section in the revised manuscript (page 21 line 23: Statistical analysis).

8) REVIEWER'S COMMENT: The authors may provide histological evidence for accurate cannula placement or injection for the micro-infusion (figure 5) and retrobeads (figure 6) experiments, respectively.

OUR RESPONSE: According to the suggestion, we showed representative figures of the cannula placement (A-C) and retrobeads injection (D-E) in Supplemental Fig. 8 in the revised manuscript.

9) REVIEWER'S COMMENT: Micro-infusion of CNO occurs near ventricles and thus, it is possible that CNO leaks into the ventricle and reaches other brain areas. Moreover, there is no evidence that this technique works in the AD. c-fos in LS, LHb and AD neurons may be sufficient to show specific postsynaptic activation by glutamatergic LHA neurons.

OUR RESPONSE: To show that micro-infusion procedure was adequately performed, we micro-infused retrobeads into the AD. As shown in Figure A in this letter, the majority of retrobeads was retained in the AD. We thus believe that CNO was appropriately delivered into the AD.

10) REVIEWER'S COMMENT (minor): The authors might consider adding a reference or two to justify the dose of CNO used.

OUR RESPONSE: Based on this suggestion, we have added the following references in the reference section in the revised manuscript. We have also modified the Methods section in the revised manuscript (page 17 lines 2-3 and lines 17-18: "CNO concentration was determined based on several reference papers").

11) REVIEWER'S COMMENT (minor): The paragraph about leptin in the introduction seems to be out of context. Is it necessary?

OUR RESPONSE: According to the suggestion, we have removed the paragraph in the revised manuscript.

12) REVIEWER'S COMMENT (minor): I think the labeling 'LS' in figure 5F is wrong.

OUR RESPONSE: Thank you for pointing it out. We have corrected the labeling in the revised manuscript.

Response to the points raised by Reviewer #2

Reviewer #2 (Remarks to the Author):

The manuscript from Fu et al investigates the neural circuits involved in hunger-related changes in the consumption of palatable and aversive stimuli. The experiments rely on an array of techniques ranging from behavioral analysis, optogenetics, chemogenetics, anatomical labeling, and viral strategies in transgenic mice. The results demonstrate that: 1) hunger increased licking for intermediated concentrations of sweet and bitter stimuli; 2) activation of AgRP-expression neurons in the arcuate nucleus of the hypothalamus recapitulates the effects of hunger; 3) specifically the effect is mediated by AgRP neurons projecting to glutamatergic neurons in the lateral hypothalamus (LHA); 4) LHA projection neurons mediate the effects of hunger on the consumption of sweets and bitters by targeting the lateral septum (LS) and the lateral habenula (LHb) respectively.

The results presented in this manuscript are interesting and very relevant as they unveil novel mechanisms responsible for modulating consumption of palatable and aversive tastants in different physiological states. The experiments are rigorously conducted and well controlled. The manuscript is clearly written and follows a linear logic. I do however have some concerns and comments on data analysis and interpretation that need to be addressed.

1) REVIEWER'S COMMENT: The results are framed as evidence for the mechanisms of hunger-induced taste modifications. However, the authors do not demonstrate that hunger affects taste sensitivity. The results of the brief access test could equally be due to changes in the perceived palatability of stimuli. The brief access test does not dissociate sensory and hedonic components. If the authors want to directly address taste sensitivity, they have to perform a taste detection task (e.g., a 2 alternative forced choice task where the animal has to discriminate different concentrations of tastants versus water). I do not think these experiments are necessary, but I do think that the authors should avoid attributing their results to changes in sensory sensitivity (hence they should edit the text accordingly). I also encourage the authors to add a discussion on the contribution of sensory (i.e., gustatory) and hedonics effects to their results.

OUR RESPONSE: To evaluate sensory aspect (the threshold of sweet taste) is regulated by LHA-projecting AgRP neurons, we performed conditioned taste aversion (CTA)

experiments that is commonly used to accurately measure perceived taste thresholds for appetitive taste qualities as mentioned in Galliard and Stratford (2016). We thus performed CTA test combined with optogenetic experiments. The results showed that optogenetic activation of LHA-projecting AgRP neurons have little impact on the dose-dependent aversive response to sucrose (Fig. 3D). These results suggest that LHA-projecting AgRP neurons did not contribute to the enhancement of sweet taste threshold, implying that they play a role in modulation of the hedonic aspect.

According to the reviewer's suggestion and these results, we have changed "sensitivity" into "preference" throughout the revised manuscript except the case of bitter taste (page 4, line 13, 17 and page 5, line 8, 15, 21, 24, page 12, line 3). This is because we agree with Reviewer 3's comment 1 of bitter taste.

We have also added a discussion on the contribution of sensory and hedonics effects to our results.

These new data have been incorporated into the Results section of the revised manuscript (page 8 line 1: "LHA-projecting AgRP neurons did not change sweet taste sensitivity"). Moreover, new figures have been added (Fig. 3 in the revised version). The procedure that we used is described in the Methods section (page 18, line 10 "Conditioned taste aversion (CTA) test for measurement of sweet taste sensitivity"). An additional discussion has been added to the Discussion section in the revised manuscript (page 12 lines 11-17 and page 13-14 lines 34-5).

2) REVIEWER'S COMMENT: Related to the point above, do the authors have any orofacial reactions data? If so, they could provide information on hunger-mediated changes in palatability. If they do not have these data, they should at least discuss the role of palatability in the discussion.

OUR RESPONSE: We do not have orofacial reaction data. We added a discussion on the role of palatability in the revised manuscript. An additional discussion has been added to the Discussion section in the revised manuscript (page 12 lines 11-17 and page 13-14 lines 34-5).

3) REVIEWER'S COMMENT: References to the literature on hunger-mediated effects on taste processing are very limited. Fundamental papers, like de Araujo et al Neuron 2006, are not cited. The authors should do a more thorough job with their citations.

OUR RESPONSE: According to the suggestion, we have added Burton *et al* 1976, de Araujo *et al* 2006, Rolls *et al* 1989 in the Reference section in the revised manuscript. We have also cited additional reference papers to enrich the Reference section in the revised manuscript.

4) REVIEWER'S COMMENT: The results in Figure 3 showing axons of LHA-projecting AgRP neurons "contacting" Vglut2positive neurons should be rigorously quantified. In the absence of a clear quantification of puncta, this evidence should be interpreted more cautiously.

OUR RESPONSE: To confirm direct synaptic connections between AgRP neurons and Vglut2^{LHA} neurons, we carried out monosynaptic rabies tracing with Vglut2-ires-Cre mice (Fig. 4B). Retrograde transsynaptic labeling from Vglut2^{LHA} neurons showed that direct synaptic connections between AgRP neurons and Vglut2^{LHA} neurons (Fig. 4C). These new data have been incorporated into the Results section of the revised manuscript (pages 8-9 lines 27-1). Moreover, new figures have been added in the revised manuscript (Fig. 4B and 4C). The procedure that we used is described in the Methods section in the revised manuscript (page 20 line 14: "Monosynaptic retrograde rabies tracing").

5) REVIEWER'S COMMENT: Statistical analyses of data from brief access tests rely on t-tests. The authors do not mention if the tests were corrected for multiple comparisons. If they did not use this correction, they should include it in the revision.

OUR RESPONSE: According to the suggestion, we have performed more appropriate statistical analyses for all data in the revised manuscripts. The data of the brief access taste test has been reanalyzed using two-way ANOVA with *post hoc* Bonferroni test. The statistical analysis that we used is described in the Methods section in the revised manuscript (page 21 line 23: Statistical analysis).

6) REVIEWER'S COMMENT: Figure 5F: the label should be LHb and not LS

OUR RESPONSE: Thank you for pointing it out. We changed the labeling in the revised manuscript.

Response to the points raised by Reviewer #3

Reviewer #3 (Remarks to the Author):

In this study, the authors set out to examine the neural substrates responsible for hunger induced taste modification. The authors demonstrate that activation of the AgRP neurons increases licking for sugar water of moderate concentration and increases the tolerance for bitter. They further show the similar behavioral changes during activation of AgRP to LH projections and direction inhibition of LH glutamatergic cells. Finally, they show that inactivation the LH projection to LS increases licking for sugar while inactivating the LH projection to LHb increases tolerance for the bitter.

1) REVIEWER'S COMMENT: The major conceptual issue is that the reviewer finds the results do not support a role of the studied cells/pathways in modulating the sensitivity to sugar taste, instead it supports a role of the cells/pathways in increasing motivation to obtain sugar. In reviewer's understanding, changes in sensitivity to the sugar taste means that the threshold to detect the sugar decreases. For example, fed animal cannot tell difference between 10mM sucrose solution from water while the starved animal can. In this study, the main difference between Agrp cell activated animals and control animals are their licking rate for moderately concentrated sugar water (e.g. 100mM). This concentration is apparently well above the detecting threshold for sugar as control animals showed clear increase in licking at that concentration in comparison to water. Thus, the increase in licking supports an increase in motivation for sugar water but not an increase in sensitivity to sugar taste. If the sensitivity to sugar is indeed increased, the reviewer will expect an increase in licking sugar water of low concentration. On the other hand, the reviewer found the claim that AgRP activation decreases bitter sensitivity is more convincing given the data in Supplementary Figure 2B. In this experiment, it does appear that AgRP activation decreased the detection threshold for the bitter as the control water-restricted animal reduced licking of water containing 0.1mM Denatonium while the AgRP activated animals did not. Also, since AgRP neurons do not drive drinking (but the cells do drive feeding and thus confound the interpretation for increased licking of sucrose water), it excludes the possibility that the increased licking is simply due to an increase in thirst.

OUR RESPONSE: Sweet substances such as sucrose are composed of both sensory and hedonic aspects (Seward 2004); thus it remains unclear whether the increased lick rate for moderate sucrose solutions during optogenetic activation of LHA-projecting AgRP

neurons is due to a change in sweet taste sensitivity or enhancement of hedonics of sweetness. To evaluate the sensory aspect of sweetness (as the threshold of sweet taste), we performed conditioned taste aversion (CTA) experiments that is commonly used to accurately measure perceived taste thresholds for appetitive taste qualities as mentioned in Galliard and Stratford (2016). To investigate the role of LHA-projecting AgRP neurons, ChR2-EYFP was introduced into AgRP neurons and optic fibers were bilaterally inserted above the LHA (Fig. 3A and 3B). After acquisition of CTA, the mice exhibited strong aversion to 300 mM sucrose solution (Fig. 3C). Importantly, optogenetic activation of LHA-projecting AgRP neurons did not affect the dose-dependent aversive response to sucrose (Fig. 3D). These results suggest that LHA-projecting AgRP neurons did not affect sweet taste threshold, implying that they play a role in modulation of the hedonic aspect of sweet taste.

Based on these results, we have changed “sensitivity” into “preference” throughout the revised manuscript except the case of bitter taste (page 4, line 13, 17 and page 5, line 8, 15, 21, 24, page 12, line 3). This is because we agree with Reviewer 3’s comment 1 of bitter taste.

We have also added a discussion on the contribution of sensory and hedonics effects to our results.

These new data have been incorporated into the Results section of the revised manuscript (page 8 line 1: "LHA-projecting AgRP neurons did not change sweet taste sensitivity"). Moreover, new figures have been added (Fig. 3 in the revised version). The procedure that we used is described in the Methods section (Page 18, line 10: “Conditioned taste aversion (CTA) test for measurement of sweet taste sensitivity”). An additional discussion has been added to the Discussion section in the revised manuscript (page 12 lines 11-17 and page 13-14 lines 34-5).

2) REVIEWER’S COMMENT: The claim that the AgRP neurons target LH glutamatergic cells need to be shown more convincingly using slice recording or monosynaptic rabies tracing. Do AgRP cells target GABAergic cells in the LH or they only target glutamatergic cells?

OUR RESPONSE: To confirm direct synaptic connections between AgRP neurons and Vglut2^{LHA} neurons, we carried out monosynaptic rabies tracing with Vglut2-ires-Cre mice. Retrograde transsynaptic labeling from Vglut2^{LHA} neurons showed that direct synaptic connections between AgRP neurons and Vglut2^{LHA} neurons exist as shown in Fig. 4B and 4C in the revised manuscript.

We next investigated the synaptic connection between AgRP neurons and GABAergic LHA neurons. Similar to the case of $Vglut2^{LHA}$ neurons, we carried out monosynaptic rabies tracing with GAD2-Cre mice that selectively express Cre in GABAergic neurons. The result indicated that parts of AgRP neurons connect to GABAergic neurons in the LHA (Supplemental Fig. 7A). We thus chemogenetically inhibited GABAergic LHA neurons to observe their effects on taste preference (Supplemental Fig. 7B). In clear contrast to the case of $Vglut2^{LHA}$ neurons, taste preference did not change by inhibition of GABAergic LHA neurons (Supplemental Fig. 7D and 7E). These results suggest that the contribution of GABAergic LHA neurons to the modulation of taste preference is much smaller than that induced by glutamatergic LHA neurons.

These new data have been incorporated into the Results section of the revised manuscript (pages 8-9 lines 27-1: "To further confirm the connection between AgRP neurons and $Vglut2^{LHA}$ neurons, we next carried out monosynaptic rabies tracing experiments..." and page 9 lines 25-26: "GABAergic neurons in the LHA are not required for the modulation of taste preference"). Moreover, new figures have been added in the revised manuscript (Figs. 4B and 4C and Supplemental Fig. 7). The procedure that we used is described in the Methods section (page 20 line 14: "Monosynaptic retrograde rabies tracing").

3) REVIEWER'S COMMENT: The paper will be significantly strengthened by showing an increased response of the AgRP cells and LH glutamatergic cells to sugar water at borderline concentration. Based on the hypothesis, one will predict that LH glutamatergic cells will not respond to low concentration sugar water in fed animal but will respond to it in fasted animals or AgRP activated animals.

OUR RESPONSE: As mentioned in OUR RESPONSE to Comment 1 of Reviewer 3, LHA glutamatergic neurons did not affect the sweet taste threshold. To further understand the physiological roles of these neuronal cells, it will be meaningful to utilize the selective expression of calcium sensors (such as GCaMP6) into the LS- or LHb- projecting $Vglut2^{LHA}$ neurons to monitor real time neuronal activities towards taste stimuli under fed or hunger conditions. We have added a discussion of this possibility in the revised manuscript (page 13 lines 9-12).

4) REVIEWER'S COMMENT: As CNO is a small molecular that is readily diffused throughout the body, local injection of CNO may affect adjacent regions. It will be better to use an intersectional retrograde labeling strategy to target the LH glutamatergic

–LHb projecting and LS projecting cells especially given that they have low overlap. It will be even more informative to record the activity of LH glutamatergic –LHb projecting and LS projecting cells to understand whether they show differential responses to sugar and bitter taste.

OUR RESPONSE: To investigate the activity of LS or LHb neurons towards sweet and bitter taste, we performed an immunohistochemical analysis of *c-fos* from the fed or fasted mice. While hunger and sweet taste cooperatively suppressed basal *c-fos* expression in the LS, bitter taste did not show such effects (Fig. 8C). By contrast, hunger selectively suppressed bitter induced *c-fos* expression in the LHb. Such suppression was not observed in the case of sweet taste (Fig. 8D). These results suggest that hunger differentially modulates taste induced *c-fos* expression in the LS and in the LHb.

These new data have been incorporated into the Results section of the revised manuscript (page 11 lines 10-11: "Fasting differentially modulates taste induced *c-fos* expression in the LS and the LHb"). Moreover, new figures have been added in the revised manuscript (Fig. 8 in the revised version). The procedure that we used is described in the Methods section in the revised manuscript (page 19 line 17: "Taste stimuli and *c-fos* experiments").

Reviewers' Comments:

Reviewer #1:

Remarks to the Author:

I previously reviewed this manuscript as reviewer #1. The authors responded well to my comments and thoroughly revised the manuscript. The quality of the manuscript is greatly improved.

I only have one additional comment about the reporting of statistics. I think that authors should provide F- and P-values of the ANOVA tests in the text or in a table.

Reviewer #2:

Remarks to the Author:

The authors have appropriately addressed my concerns by adding new experiments and analyses, and by editing the text. As a result, the manuscript, which was already very interesting, is now stronger.

Reviewer #3:

Remarks to the Author:

The authors have addressed all my questions. I now support its publication in Nature Communication.

August 23, 2019

Ken-ichiro Nakajima, PhD
National Institute for Physiological Sciences
Division of Endocrinology and Metabolism
Okazaki, Aichi 4448585 Japan

Response to the Referees' comments

Nature Communications manuscript: NCOMMS-18-24432B

TITLE: Hypothalamic neuronal circuits regulating hunger-induced taste modification
by O. Fu *et al.*

We would like to thank the three reviewers for their constructive comments and suggestions. We appreciate their time and effort and believe that this manuscript has been significantly improved based upon their suggestions.

Our point-by-point response to the comment of the reviewer 1 is detailed in the attached letter.

We are confident that our response letter adequately address the reviewers' concerns and hope that the manuscript in its present form is now acceptable for publication in *Nature Communications*.

Thank you very much for your consideration.

Sincerely,

Ken-ichiro Nakajima, PhD

Response to the point raised by Reviewer #1

Reviewer #1 (Remarks to the Author):

I previously reviewed this manuscript as reviewer #1. The authors responded well to my comments and thoroughly revised the manuscript. The quality of the manuscript is greatly improved.

1) REVIEWER'S COMMENT: I only have one additional comment about the reporting of statistics. I think that authors should provide F- and P-values of the ANOVA tests in the text or in a table.

OUR RESPONSE: According to the suggestion, we have added F- and P-values of the ANOVA tests (red) in the main and supplementary figure legends as follows.

Figure legends (main figures)

Fig. 1. Chemogenetic activation of AgRP neurons induces changes in taste preference. (A) Schematic image of the brief access taste test. The number of licks is measured during 10 s from the first lick. (B and C) Sweet (B) or bitter (C) taste preferences in fed or fasted mice. Sucrose or denatonium-sucrose solutions were presented to fed or 23 h fasted C57BL/6J WT mice. $n=6$, $F=17.81$, $P=9.4\times 10^{-5}$ in (B) and $n=6$, $F=4.14$, $P=0.045$ in (C), two-way ANOVA with Bonferroni *post hoc* test. (D) Bilateral injection of AAV encoding Cre-dependent hM3Dq-mCherry or hM4Di-mCherry into the arcuate nucleus (ARC) of AgRP-ires-Cre mouse. (E) Representative image showing hM3Dq-mCherry-expressing AgRP neurons (left) in the AgRP-hM3Dq mouse and hM4Di-mCherry-expressing AgRP neurons (right) in the AgRP-hM4Di mouse. (F) Chemogenetic activation of AgRP neurons led to acute food intake in AgRP-hM3Dq mice during the light period. $n=6$, paired Student's *t*-test. (G and H) Brief access taste tests for sweet (G) or bitter (H) measured in AgRP-hM3Dq mice treated with saline or CNO (1.0 mg/kg i.p) during the light cycle. $n=6$, $F=8.783$, $P=0.0045$ in (G) and $n=6$, $F=7.929$, $P=0.0064$ in (H), two-way ANOVA with Bonferroni *post hoc* test. (I) Chemogenetic inhibition of AgRP neurons led to a reduction of food intake in AgRP-hM4Di mice during the dark cycle. $n=7$, paired Student's *t*-test. (J and K) Brief access taste tests for sweet (J) or bitter (K) measured in AgRP-hM4Di mice treated with saline or CNO (1.0 mg/kg i.p) during the dark cycle. $n=7$, $F=4.748$, $P=0.032$ in (J) and $n=7$, $F=1.84$, $P=0.142$ in (K), two-way ANOVA with Bonferroni *post hoc* test. The experiments were carried out with

8- to 16-week-old male mice. Data are given as means \pm SEM. * P <0.05, ** P <0.01, *** P <0.001.

Fig. 2. Optogenetic activation of LHA-projecting AgRP neurons modulates preference for sweet and bitter tastes.

(A) AAV-FLEX-rev-ChR2-tdTomato was bilaterally injected into the ARC of AgRP-ires-Cre mice and optical fibers were placed above the projection regions of AgRP neurons. (B) Schematic image of the brief access taste test during *in vivo* optogenetic activation of AgRP axon terminals in AgRP-ChR2 mice. (C–E) Brief access taste tests towards sweet (D) or bitter (E) solutions during photostimulation of the soma of AgRP^{ARC} neurons. $n=11$, $F=19.41$, $P=2.8\times 10^{-5}$ in (D) and $n=11$, $F=6.926$, $P=0.01$ in (E), two-way ANOVA with Bonferroni *post hoc* test. (F–N) Brief access taste tests for sweet and bitter solutions when exclusively activating PVH-projecting (F–H), LHA-projecting (I–K), and CEA-projecting (L–N) AgRP neurons, respectively. $n=6$, $F=2.263$, $P=0.138$ in (G), $n=6$, $F=0.2445$, $P=0.87$ in (H), $n=5$, $F=11.42$, $P=0.0015$ in (J), $n=5$, $F=13.88$, $P=0.0005$ in (K), $n=5$, $F=3.174$, $P=0.081$ in (M), $n=5$, $F=1.729$, $P=0.194$ in (N), two-way ANOVA with Bonferroni *post hoc* test. All experiments were carried out with 10- to 16-week-old male mice. Data are given as means \pm SEM. * P <0.05, ** P <0.01, *** P <0.001, as compared with the corresponding control group.

Fig. 3. LHA-projecting AgRP neurons do not affect sweet taste sensitivity after CTA.

(A) Schematic image of the injection of AAV-DIO-ChR2-EYFP into the ARC with bilateral optic fiber insertion above the LHA for photostimulation in the AgRP-ChR2-EYFP mouse. (B) Representative image showing ChR2-EYFP expression at axon terminals of AgRP neurons in the LHA and the approximate placement of an optic fiber (dashed lines) (C) Licks for 300 mM sucrose in 10 s before and after CTA conditioning. $n=6$, paired Student's *t*-test. (D) Brief access taste tests for sucrose solution in the presence or absence of optogenetic activation of LHA-projecting AgRP neurons after CTA conditioning. $n=6$, $F=1.272$, $P=0.262$, two-way ANOVA with Bonferroni *post hoc* test. All experiments were carried out with 10- to 16-week-old male mice. Data are given as means \pm SEM. *** P <0.001.

Fig. 5. Vglut2 neurons are necessary for hunger-induced modulation of taste preference.

(A) Bilateral injection of AAV-DIO-hM4Di-mCherry or AAV-DIO-hM3Dq-mCherry into the LHA of Vglut2-ires-Cre mice. (B–C) Representative images of hM4Di-mCherry (B) and hM3Dq-mCherry (C) expression in Vglut2^{LHA} neurons. Fx, fornix. (D)

Chemogenetic inhibition (CNO 1.0 mg/kg i.p.) of Vglut2 neurons in the LHA promotes food intake in Vglut2-hM4Di mice within 1 h. $n=7$, paired Student's *t*-test. (E–F) Brief access tests with sweet (E) and bitter (F) taste solutions in Vglut2^{LHA}-hM4Di mice in the presence or absence of CNO (1.0 mg/kg i.p.). $n=6$, $F=11.99$, $P=0.001$ in (E) and $n=6$, $F=5.37$, $P=0.023$ in (F), two-way ANOVA with Bonferroni *post hoc* test. (G) Chemogenetic activation (CNO 1.0 mg/kg i.p.) of Vglut2 neurons in the LHA led to a decrease in 1 h food intake in 23 h-fasted Vglut2^{LHA}-hM3Dq mice. (H–I) Brief access test with sweet (H) and bitter (I) taste solutions in Vglut2^{LHA}-hM3Dq mice under fed or 23 hr-fasted conditions in the presence or absence of CNO (1.0 mg/kg i.p.). $n=6$, $F=21.77$, $P=1.5 \times 10^{-8}$ in (H) and $n=6$, $F=19.51$, $P=1.1 \times 10^{-7}$ in (I), two-way ANOVA with Bonferroni *post hoc* test. All experiments were carried out with 10- to 16-week-old male mice. Data are given as means \pm SEM. $*P<0.05$, $**P<0.01$, $***P<0.001$, as compared to the saline group (D–G) and to the fasted (saline) group (H–I).

Fig. 6. Two distinct hypothalamic pathways contribute to the modulation of sweet and bitter preference.

(A) Injection of AAV-DIO-hM4Di-mCherry into the LHA of Vglut2-ires-Cre mice. (B) Representative image of the projection regions (LS, AD, and LHb) of Vglut2^{LHA} neurons. (C–K) Brief access taste tests after local inhibition of Vglut2^{LHA} neurons projecting to the LS (C), LHb (F), or AD (I) by microinfusion of CNO, respectively. Preferences towards sweet taste (D, G, J) or bitter taste (E, H, K) were evaluated 10 min after microinjection of CNO (0.1 mg/ml, 200 nl). $n=6$, $F=34.32$, $P=8.9 \times 10^{-8}$ in (D), $n=6$, $F=7.526$, $P=0.0077$ in (E), $n=6$, $F=10.59$, $P=0.0016$ in (G), $n=6$, $F=10.72$, $P=0.0016$ in (H), $n=5$, $F=3.564$, $P=0.063$ in (J), $n=5$, $F=2.823$, $P=0.098$ in (K), two-way ANOVA with Bonferroni *post hoc* test. All experiments were carried out with 10- to 16-week-old male mice. Data are given as means \pm SEM. $*P<0.05$, $**P<0.01$, $***P<0.001$.

Fig. 8. Fasting differentially affects taste induced *c-fos* expression in the LS and the LHb.

(A and B) Representative images showing *c-fos* positive cells in the LS (A) and in the LHb (B) (C and D) Quantification of *c-fos* positive cells in the LS (C) and in the LHb (D). The mice were treated with sucrose or bitter taste under fed or fasted conditions. $n=3–5$ mice per group and approximately 10 brain slices per mouse were analyzed. $F=51.25$ $P=4.9 \times 10^{-8}$ in (C), $F=11.1$ $P=0.0069$ in (D), one-way ANOVA with Dunnett's *post hoc* test. All experiments were carried out with 10- to 16-week-old male mice. Data

are given as means \pm SEM. * P <0.05, ** P <0.01, *** P <0.001 as compared with the control group.

Supplementary Figure legends

Supplementary Fig. 1. CNO induce *c-fos* expression in AgRP-hM3Dq-mCherry mice but has no effect on *c-fos* expression, food intake, and taste preference in AgRP-ires-Cre mice injected with the control AAV encoding Cre-dependent mCherry.

(A) *c-fos* expression in the ARC of AgRP-hM3Dq-mCherry mice 1 h after CNO injection (1.0 mg/kg i.p.). (B) AgRP-ires-Cre mice were injected with AAV-DIO-mCherry (AgRP-mCherry mice). Representative immunohistochemical image shows little *c-Fos* expression in the ARC of AgRP-mCherry mice 1 h after CNO injection (1.0 mg/kg i.p.). (C) Food intake in AgRP-mCherry mice. Acute feeding was not triggered by CNO treatment (1.0 mg/kg i.p.) in AgRP-mCherry mice. $n=7$, paired Student's *t*-test. (D) No difference was observed in preference for sucrose taste in AgRP-mCherry mice treated with CNO (1.0 mg/kg i.p.). $n=7$, $F=0.003$, $P=0.9565$, two-way ANOVA with Bonferroni *post hoc* test. All experiments were carried out with 10- to 16- week-old male mice. Data are given as means \pm SEM.

Supplementary Fig. 2. Chemogenetic activation of AgRP neurons also modifies taste preference towards sucralose and citric acid.

(A) Taste preference toward the non-calorie sweetener, sucralose, in AgRP-hM3Dq mice treated with saline or CNO (1.0 mg/kg i.p.). $n=6$, $F=5.6$, $P=0.021$, two-way ANOVA with Bonferroni *post hoc* test. (B) Taste preference toward denatonium in 23 h water deprived AgRP-hM3Dq mice treated with saline or CNO (1.0 mg/kg i.p.). $n=6$, $F=7.98$, $P=0.0064$, two-way ANOVA with Bonferroni *post hoc* test. (C) Taste preference toward citric acid in saline-treated fed AgRP-hM3Dq mice (black), CNO-treated (1.0 mg/kg i.p.) fed AgRP-hM3Dq mice (red), or overnight-fasted AgRP-hM3Dq mice (blue). $n=8$, $F=7.45$, $P=0.0009$, two-way ANOVA with Bonferroni *post hoc* test. All experiments were carried out with 10- to 16-week-old male mice. Data are given as means \pm SEM. * P <0.05, ** P <0.01, *** P <0.001 as compared to saline group.

Supplementary Fig. 3. Chorda tympani nerve responses to gustatory stimuli are unaffected by chemogenetic activation of AgRP neurons in AgRP-hM3Dq mice. Chorda tympani nerve response (relative to 100 mM NH_4Cl) for sucrose (A), sucralose (B), and denatonium (C) in AgRP-hM3Dq mice after either saline or CNO treatment. Mice were anesthetized using pentobarbital and nerve recording to taste solution was

performed for 10 min after single injection of saline or CNO (1.0 mg/kg i.p.). Taste sensitivities for sweet and bitter tastants were not affected by the chemogenetic activation of AgRP neurons. All experiments were carried out with 10- to 20- week-old male mice (n=5–6 per group). Data are given as means \pm SEM.

Supplementary Fig. 5. Brief access taste test for sucralose and citric acid solution when optogenetically activating the ARC (soma) (A–C), PVH-projecting (D–F), LHA-projecting (G–I), and CEA-projecting AgRP (J–L) neurons in AgRP-ChR2 mice. n=6, $F=4.113$, $P=0.0496$ in (B), n=6, $F=15.59$, $P=0.0002$ in (C), n=6, $F=0.165$, $P=0.686$ in (E), n=6, $F=2.836$, $P=0.098$ in (F), n=5, $F=16.27$, $P=0.0002$ in (H), n=5, $F=16.69$, $P=0.0001$ in (I), n=5, $F=0.03$, $P=0.8637$ in (K), n=5, $F=1.899$, $P=0.174$ in (L), two-way ANOVA with Bonferroni *post hoc* test. All experiments were carried out with 10- to 16-week-old male mice. Data are given as means \pm SEM. * $P<0.05$, ** $P<0.01$.

Supplementary Fig. 6. Selective activation of PVT-projecting or BNST-projecting AgRP neurons does not lead to a change in taste preference in AgRP-ChR2 mice. (A) Representative image of the optic fiber implantation in the PVT. (B–D) 1h food intake (B), sucrose taste preference (C), and bitter taste preference (D) when optogenetically activating PVT-projecting AgRP neurons in AgRP-ChR2 mice. n=6, paired Student's *t*-test in (B). n=6, $F=1.716$, $P=0.196$ in (C) and n=6, $F=0.019$, $P=0.891$ in (D), two-way ANOVA with Bonferroni *post hoc* test. (E) Representative image of the optic fiber implantation in the BNST. (F–H) 1 h food intake (F), sucrose taste preference (G), and bitter taste preference (H) when optogenetically activating BNST-projecting AgRP neurons in AgRP-ChR2 mice. n=6, paired Student's *t*-test in (F). n=7, $F=0.016$, $P=0.898$ in (G) and n=7, $F=0.102$, $P=0.75$ in (H), two-way ANOVA with Bonferroni *post hoc* test. All experiments were carried out with 8- to 16- week-old male mice. Data are given as means \pm SEM.

Supplementary Fig. 7 GABAergic neurons in the LHA connect to AgRP neurons but are not necessary for hunger-induced taste modification.

(A) Monosynaptic rabies tracing of GABAergic neurons in the LHA. Enlarged view shows parts of AgRP neurons (red) monosynaptically connected to GABAergic LHA neurons. (B) Bilateral injection of AAV encoding Cre-dependent hM4Di-mCherry into the LHA of *Vgat-ires-Cre* mouse (left) and a representative image of expression of hM4Di-mCherry in LHA (right). Fx, fornix. (C) Chemogenetic inhibition (CNO 1.0 mg/kg i.p.) of *Vgat* neurons in the LHA suppressed food intake in *Vgat-hM4Di* mice

within 2 h from the beginning of the dark cycle. n=6, paired Student's *t*-test. (D and E) Brief access taste tests with sweet (D) and bitter (E) taste solutions in Vgat-hM4Di mice in the presence or absence of CNO (1.0 mg/kg i.p). n=6, $F=0.311$, $P=0.579$ in (D) and n=6, $F=3.331$, $P=0.072$ in (E), two-way ANOVA with Bonferroni *post hoc* test. Data are given as means \pm SEM. * $P<0.05$.